# Perturbation of base excision repair sensitizes breast cancer cells to APOBEC3 deaminase-mediated mutations

**Birong Shen\*, Joseph H Chapman, Michael F Custance, Gianna M Tricola, Charles E Jones, Anthony V Furano\***

Section on Genomic Structure and Function, Laboratory of Cell and Molecular Biology, National Institute of Diabetes and Digestive and Kidney Disease, National Institutes of Health, Bethesda, United States

**Abstract** Abundant APOBEC3 (A3) deaminase-mediated mutations can dominate the mutational landscape ('mutator phenotype') of some cancers, however, the basis of this sporadic vulnerability is unknown. We show here that elevated expression of the bifunctional DNA glycosylase, NEIL2, sensitizes breast cancer cells to A3B-mediated mutations and double-strand breaks (DSBs) by perturbing canonical base excision repair (BER). NEIL2 usurps the canonical lyase, APE1, at abasic sites in a purified BER system, rendering them poor substrates for polymerase β. However, the nicked NEIL2 product can serve as an entry site for Exo1 in vitro to generate single-stranded DNA, which would be susceptible to both A3B and DSBs. As NEIL2 or Exo1 depletion mitigates the DNA damage caused by A3B expression, we suggest that aberrant NEIL2 expression can explain certain instances of A3B-mediated mutations.

**\*For correspondence:**
birong.shen@nih.gov (BS);
avf@helix.nih.gov (AVF)

**Competing interests:** The authors declare that no competing interests exist.

## Introduction

The APOBEC3 (A3) family of single-stranded cytidine deaminases, are members of the AID/apolipoprotein B mRNA editing enzyme, catalytic peptide-like (APOBEC) deaminases (*Conticello, 2008*). These enzymes are involved in important physiological functions such as somatic hypermutation (*Peled et al., 2008*) and defense against retroelements (*Bishop et al., 2004*; *Bogerd et al., 2006*; *Refsland and Harris, 2013*). However, those with a preference for the cytosine (C) of TpC, notably A3A, A3B, A3C, A3F, A3H-1 can contribute to the repertoire of extensive somatic mutations ('mutator phenotype') (*Bielas et al., 2006*) that typify many cancers including the distinctive, strand-coordinated clusters of mutations termed 'kataegis' (*Alexandrov et al., 2013a*; *Burns et al., 2013a*; *Burns et al., 2013b*; *Chan and Gordenin, 2015*; *Helleday et al., 2014*; *Nik-Zainal et al., 2012*; *Nik-Zainal et al., 2016*; *Petljak et al., 2019*; *Roberts and Gordenin, 2014*; *Roberts et al., 2013*; *Starrett et al., 2016*; *Taylor et al., 2013*). In addition to the preference for single-stranded TpC, generation of approximately equal amounts of transitions (T) and transversions (A or G) provides a distinctive signature for A3-mediated mutations, which have been found in transiently unpaired regions of DNA that arise during DNA replication, recombination, and transcription (*Haradhvala et al., 2016*; *Hoopes et al., 2016*; *Morganella et al., 2016*; *Roberts et al., 2012*; *Seplyarskiy et al., 2016*).

Although A3B is a major source of A3 mutations in some cancers (*Burns et al., 2013a*; *Burns et al., 2013b*) and can be overexpressed as a function of cell proliferation in breast cancers (*Cescon et al., 2015*), the frequency of A3B mutations is not always correlated with A3B expression (*Cescon et al., 2015*; *Nik-Zainal et al., 2014*; *Roberts and Gordenin, 2014*). Furthermore, the aforementioned single-stranded A3 substrates are also present in non-cancerous cells, and A3 expression is rather ubiquitous (*Refsland et al., 2010*). Therefore, a major issue is why some cancers

become sensitized to the activity of APOBEC enzymes and whether it is related to dysregulation of DNA repair.

We had earlier shown that the repair of plasmid-borne mismatches can induce flanking A3-mediated mutations in HeLa cells. Although the mismatches (*e.g.*, U/G) were invariably repaired by base excision repair (BER), this process was sometimes hijacked by non-canonical mismatch repair (MMR), which generates single-stranded APOBEC substrates (*Chen et al., 2014*). In its simplest form, BER consists of a concerted series of reactions: removal of U by a Uracil-DNA Glycosylase (UDG, UNG in mammals) to generate an abasic (AP) site; scission of the AP site by apurinic/apyrimidinic endonuclease 1 (APE1) to generate a nick with a 3'OH and a 5' deoxyribophosphate (5'dRP); insertion of the complementary base and removal of the 5'dRP by polymerase β (Polβ); ligation of the nick by DNA Ligase I or Ligase III-XRCC1 (*Beard et al., 2006*; *Robertson et al., 2009*). Because some of these intermediates can be highly mutagenic, these reactions are tightly coupled and sequestered (*Fu et al., 2012*; *Prasad et al., 2010*). Nonetheless, under certain conditions MMR can access the 3'OH terminated nick, leading to an Exo1-mediated resection that exposes single-stranded DNA opposite to the nicked strand (*Kadyrov et al., 2006*; *Peña-Diaz et al., 2012*; *Pluciennik et al., 2010*; *Schanz et al., 2009*). Given that thousands of U/G mismatches and AP sites are produced daily (*Atamna et al., 2000*; *Barnes and Lindahl, 2004*; *Frederico et al., 1993*), perturbations that compromise the integrity of the BER complex could lead to DNA damage (*Fu et al., 2012*).

Here we demonstrate in breast cancer cell lines that elevated expression of NEIL2, a bifunctional glycosylase normally involved in oxidative base excision repair (*Chakraborty et al., 2015*; *Das et al., 2006*; *Hazra et al., 2002*; *Wiederhold et al., 2004*), facilitates A3B-mediated mutations during U/G mismatch repair and induces double-strand breaks (DSBs) in genomic DNA. We further show that purified NEIL2 disrupts canonical BER by outcompeting APE1 for AP sites, thereby providing a possible mechanistic explanation for how this instance of DNA repair dysregulation contributes to the mutational landscape in breast cancer cells.

## Results

### A3B activity is not the only determinant of repair-induced mutations

To examine U/G mismatch repair-induced effects in breast cancer cell lines, we transfected shuttle vectors containing no or a U/G mismatch into four established breast cancer cell lines: MCF7, HCC1569, Hs578T and MDA-MB-453, and screened for mutations in the reporter region that flanked the U/G. The reporter region consists of the *E. coli* SupF gene and its promoter on the shuttle vector pSP189-SnA (*Figure 1A* and *Figure 1—figure supplement 1A*). Inactivating mutations of the SupF region induced by U/G repair cannot suppress the mutated β galactosidase gene in the MBM7070 *E. coli* strain, resulting in white colonies on the indicator plates (*Figure 1A*, bottom row). U/G-repair did not induce mutations in MDA-MB-453, but it did so in Hs578T (*Figure 1B*, bottom bar graph), despite similar levels of A3B transcripts (*Figure 1B*, upper bar graph) and comparable nuclear TC-specific deaminase activity (*Figure 1C* and *Figure 1—figure supplement 1B,C*) in these cell lines. The discrepancy between statistically significant amounts of repair-induced mutations and A3B expression also occurred in other cell lines (*Figure 1B*). We sequenced the mutated reporter regions of plasmids from all the white colonies, and essentially all of the repair-induced mutations in Hs578T and HCC1569 exhibited an A3 signature, displayed here on the complement of the TC-containing strand – thus, G was the most frequently mutated nucleotide and >70% of mutated bases in Hs578T cells and >50% in HCC1569 cells involved A<u>G</u>A, C<u>G</u>A, or T<u>G</u>A (*Figure 1D,E* and *Figure 1—figure supplement 1D*).

Among the seven A3 enzymes, A3A, A3B, A3C, and A3H localize to the nucleus and prefer TC sites (*Lackey et al., 2013*). Quantitative real-time PCR (qRT-PCR) showed that only A3B and A3C were expressed in Hs578T cells (*Figure 1—figure supplement 2A*). Knockdown of A3B by siRNA (*Figure 1—figure supplement 2B*, left bar graph) reduced both in vitro deaminase activity (*Figure 1—figure supplement 2C*) and the extent of U/G repair-induced mutation rate in Hs578T cells (*Figure 1F*), implicating the role of A3B in repair-induced mutagenesis. However, knockdown of A3C (*Figure 1—figure supplement 2B*, right bar graph) did not affect either in vitro deaminase activity (*Figure 1—figure supplement 2C*) or U/G repair-induced mutation (*Figure 1—figure supplement 2D*), indicating that A3B was the major deaminase activity involved in the repair-induced

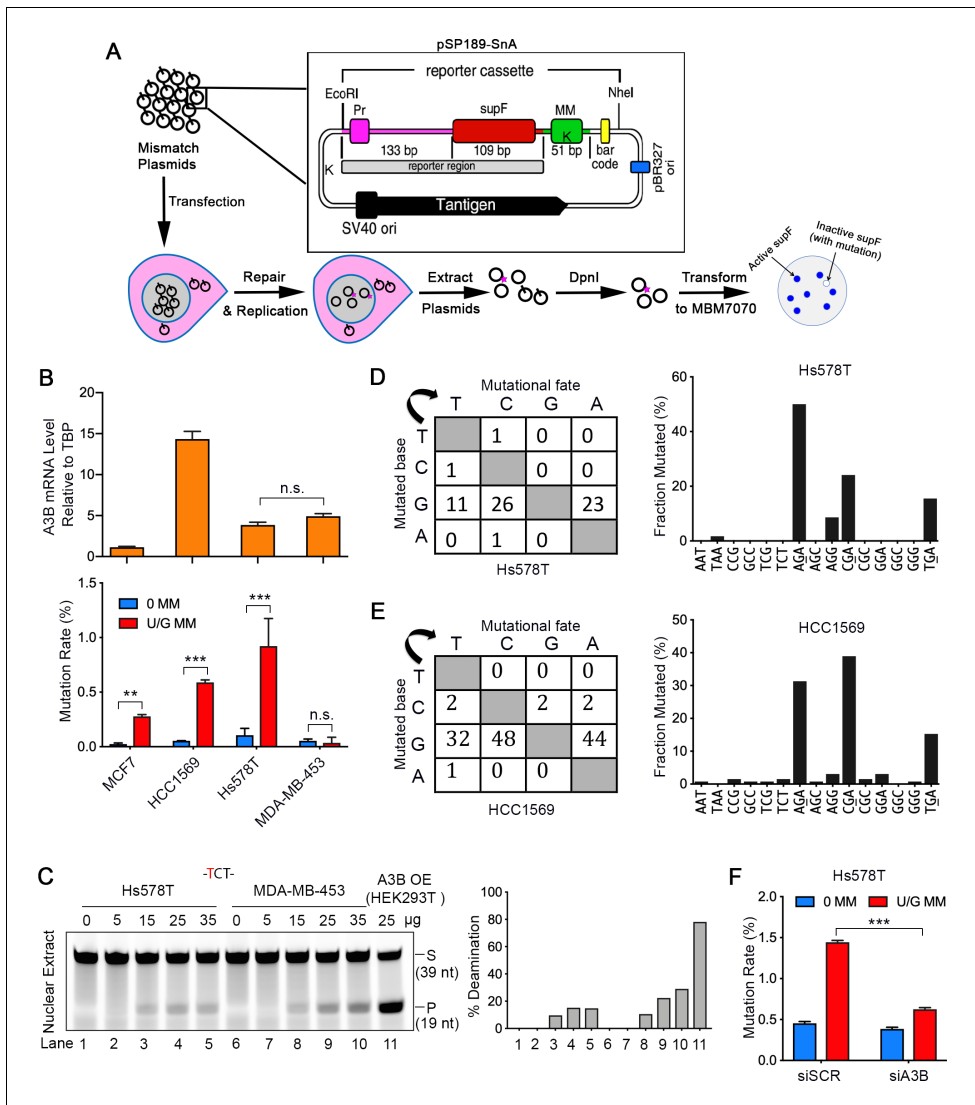

**Figure 1.** A3B activity is not the only determinant of repair-induced mutations. (**A**) Schematic depicting the shuttle vector assay to detect U/G MM repair-induced mutations. MM, no mismatch or U/G mismatch. K depicts location of KpnI site. (**B**) Upper panel: qRT-PCR of A3B relative to the housekeeping gene TBP. Lower panel: mutation rate (scored as % of white/total colonies) induced by U/G mismatch repair in MCF7, HCC1569, Hs578T, and MDA-MB-453 breast cancer cell lines. 0 MM, no mismatch; U/G MM, U/G mismatch. Error bars represent s.d., n = 2 for MCF7, HCC1569 and MDA-MB-453 cells; n = 5 for Hs578T cells. [**]P < 0.01; [***]P < 0.001; n.s., no significant difference by two-tailed unpaired Student's t test. (**C**) Concentration gradient of in vitro deaminase assay using nuclear extracts from Hs578T and MDA-MB-453 cells against a -TCT-containing fluorescein-labeled single strand oligonucleotide (39 nt). The amounts of total protein used are listed on top of the gel. The right panel shows quantification of the deamination percentage. The deamination activity is specific for -TCT- (Figure 1—figure supplement 1B). The time course deamination is shown in Figure 1—figure supplement 1C. S, substrate; P, product. (**D and E**) Mutation matrices and 5'-Trinucleotide context of mutations induced by U/G MM repair in Hs578T (**D**) and HCC1569 (**E**) cells. C is the most frequently mutated base and 70% of the mutated bases are in a 5'-GA (reverse complement of 5'-TC) motif. (**F**) A3B deficiency decreases U/G mismatch repair-induced mutagenesis. 0 MM, no mismatch; U/G MM, U/G mismatch. Error bars represent s.d., n = 3. [***]P < 0.001 by two-tailed unpaired Student's t test.

The online version of this article includes the following figure supplement(s) for figure 1:

**Figure supplement 1.** Shuttle vector-based assay of repair-induced mutations and A3 deaminase activity in breast cancer cell lines.

**Figure supplement 2.** A3B, but not A3C, is correlated with the repair-induced mutagenesis.

mutations in Hs578T cells. That non-mutagenic MDA-MB-453 cells contained a similar level of C-deaminase activity as the mutagenic Hs578T cells (*Figure 1C* and *Figure 1—figure supplement 1C*) indicates that A3 activity *per se* is not sufficient to cause repair-induced mutations.

## NEIL2 facilitates repair-induced mutagenesis

To explore whether dysregulation of DNA repair sensitized cells to A3B-mediated mutations, we compared the expression of 84 DNA repair enzymes in Hs578T and MDA-MB-453 cells using $RT^2$ Profiler PCR array. Consistent with previous RNA-seq data (*Klijn et al., 2015*), most of the tested genes were expressed at lower levels in Hs578T than in MDA-MB-453 (*Figure 2—source data 1*). However, two genes, NEIL2 and TREX1 were significantly upregulated (*Figure 2A*). TREX1 is a 3′−5′ exonuclease that degrades cytosolic single and double-stranded DNA, which can illicit an inflammatory innate immune response (*Crow et al., 2006*). NEIL2 is a bifunctional glycosylase involved in BER of oxidized bases (*Chakraborty et al., 2015*; *Hazra et al., 2002*; *Mandal et al., 2012*; *Wiederhold et al., 2004*) and methyl-cytosine demethylation (*Schomacher et al., 2016*). NEIL2 catalyzes both base removal and scission of the ensuing AP sites but leaves a 3′-phosphate (3′P) that is removed by PNKP (a polynucleotide 3′kinase and phosphatase) to generate a 3′OH that primes Polβ (*Das et al., 2006*; *Wiederhold et al., 2004*).

To determine whether either gene was involved in repair-induced mutations, we depleted NEIL2 or TREX1 by siRNA in Hs578T cells (*Figure 2—figure supplement 1A,B*). While NEIL2 knockdown reduced the U/G repair-induced mutation rate by ~70% (*Figure 2B*), knockdown of TREX1 had no effect (*Figure 2—figure supplement 1C*). To corroborate the NEIL2 siRNA results, we packaged lentivirus expressing NEIL2 shRNA (#1 targets NEIL2 3′UTR; #2 targets NEIL2 ORF) to generate NEIL2-stable-knockdown Hs578T cell lines (shNEIL2#1 and shNEIL2#2, *Figure 2C*). Consistent with the siRNA knockdown result, repair-induced mutations were reduced in both NEIL2-depleted cell lines (*Figure 2D*). Moreover, restoring NEIL2 rescued U/G repair-induced mutations (*Figure 2E*). And finally, overexpressing NEIL2 in the low-NEIL2 expression cell line, MDA-MB-453, doubled the U/G repair-induced mutation rate (*Figure 2F*). Taken together, these results indicate that NEIL2 facilitates U/G repair-induced mutations.

In addition, we found a positive correlation between U/G repair-induced mutations and the relative NEIL2 expression (qRT-PCR data) in the four breast cancer cell lines (*Figure 2—figure supplement 2A,B*). These results are consistent with the involvement of NEIL2 in the APOBEC-mediated mutations.

## NEIL2 participates in A3B-mediated genomic DNA damage

The above results indicate that the elevated level of NEIL2 in Hs578T cells sensitizes them to the single strand deaminase activity of A3B during DNA repair. Single-stranded DNA is prone to further damage including DSBs, which can be detected as γH2AX foci (*Bonner et al., 2008*; *Burns et al., 2013a*; *Landry et al., 2011*; *Morel et al., 2017*; ). Therefore, we used γH2AX foci as a proxy for NEIL2-facilitated, A3B-induced genomic damage. Expression of exogenous A3B generated a statistically significant increase in γH2AX foci, which was markedly decreased in NEIL2-depleted Hs578T cells (*Figure 3A*). This was not the result of diminished deaminase activity as A3B deaminase activity generated from the A3B-HA expression vector was unaffected in NEIL2-depleted cell lines (*Figure 3B* and *Figure 3—figure supplement 1A*). Rescue of NEIL2 with exogenous NEIL2-HA (*Figure 3—figure supplement 1B*) restored A3B-induced γH2AX foci in NEIL2-depleted cells (*Figure 3C* and *Figure 3—figure supplement 1C*). Furthermore, similar to U/G repair-induced mutagenesis (*Figure 2F*), expression of NEIL2-HA in non-mutagenic MDA-MB-453 cells increased A3B-mediated γH2AX foci (*Figure 3D*). These results indicate that NEIL2 is involved in A3B-induced genomic DNA damage.

## NEIL2 outcompetes APE1 at AP sites

The specificity of NEIL2 for single-stranded oxidized bases predicts that NEIL2 would not directly participate in the repair of the introduced U/G mismatch. However, our results demonstrated that NEIL2 interacts with U/G repair and increases susceptibility to A3B-mediated mutations and DNA damage (*Figures 1* and *2*). To gain mechanistic insight into this process we purified human His-tagged NEIL2 protein (*Figure 4—figure supplement 1A*). Consistent with previous studies

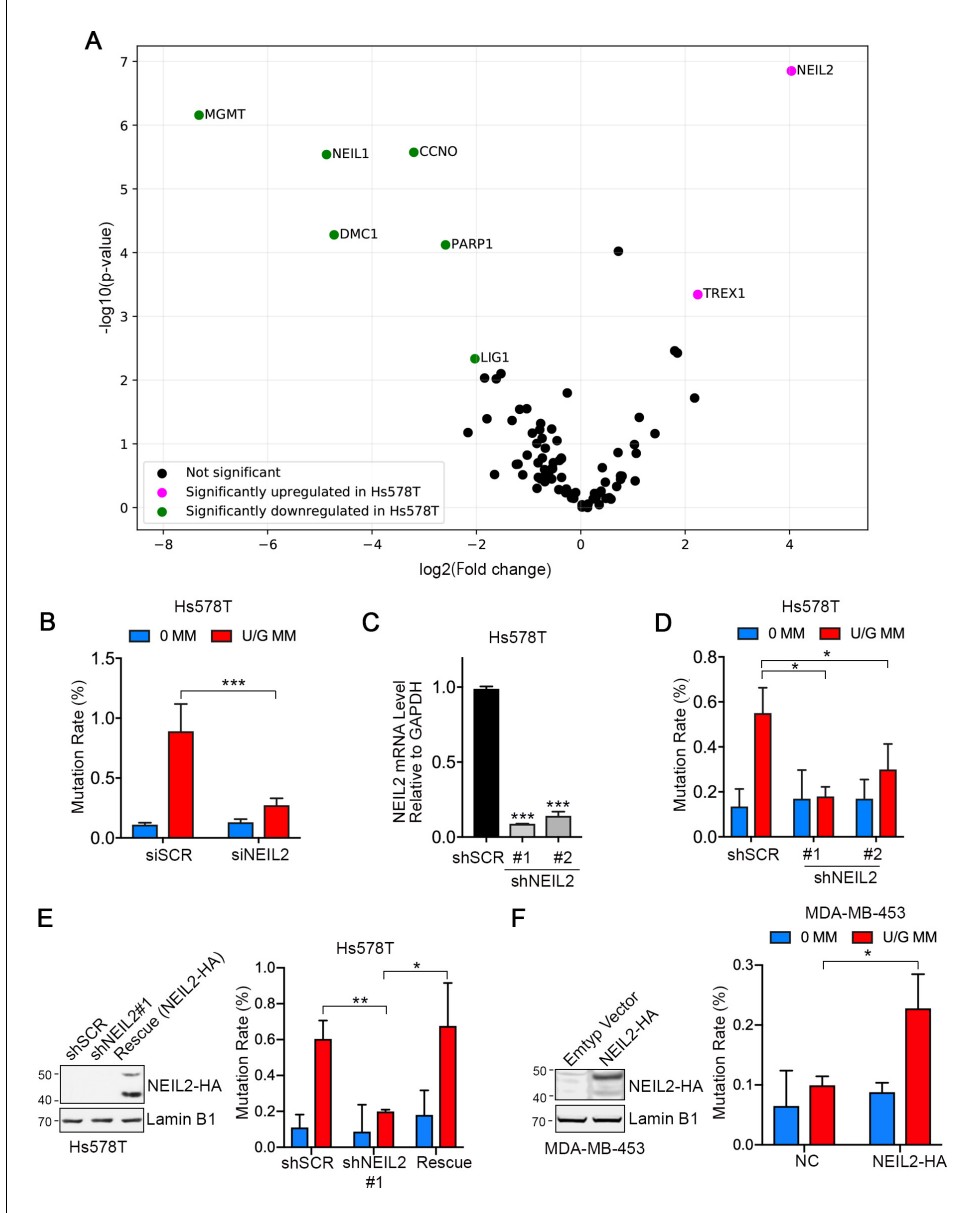

**Figure 2.** NEIL2 facilitates repair-induced mutagenesis. (**A**) Volcano plot of transcription levels of 84 DNA repair enzymes determined by RT$^2$ Profiler PCR Array for Hs578T cells (high mutation rate cell line) relative to MDA-MB-453 cells (low mutation rate cell line). Genes significantly downregulated and upregulated in Hs578T cells are highlighted in green and magenta, respectively. Data were generated from four independent determinations (*Figure 2—source data 1*). (**B**) U/G repair-induced mutation using the shuttle vector assay upon NEIL2 knockdown by siRNA in Hs578T cells. NEIL2 depletion decreased U/G MM repair-induced mutagenesis. siSCR, scramble siRNA; 0 MM, no mismatch; U/G MM, U/G mismatch. Error bar represents s.d., n = 3. ***P < 0.001 by two-tailed unpaired Student's t test. (**C**) qRT-PCR shows knockdown efficiency of NEIL2 relative to GAPDH in NEIL2-stable-knockdown Hs578T cell lines (shNEIL2#1 and shNEIL2#2). shSCR, scramble shRNA; shNEL2#1 targets NEIL2 3'UTR; shNEIL2#2 targets NEIL2 ORF. Error bars represent s.d., n = 3. ***P < 0.001 by two-tailed unpaired Student's t test. (**D**) U/G repair-induced mutation using the shuttle vector assay in NEIL2-stable-knockdown Hs578T cell line. NEIL2 depletion decreased U/G MM repair-induced mutagenesis. 0 MM, no mismatch; U/G MM, U/G mismatch. Error bars represent s.d., n = 3. *P < 0.05 by two-tailed unpaired Student's t test. (**E**) Rescue of NEIL2 by NEIL2-HA overexpression vector restores U/G mismatch repair-induced mutation rate in NEIL2-stable-knockdown Hs578T cell line shNEIL2#1 (targets NEIL2 3'UTR). Western blot (left panel) shows NEIL2-HA overexpression for rescue of NEIL2 in shNEIL2-#1 Hs578T cell line. Lamin B1 serves as a loading control. Error bars represent s.d., n = 3. *P < 0.05; **P < 0.01 by two-tailed unpaired Student's t test. (**F**) Overexpression of NEIL2-HA (pPM-NEIL2-3'HA) in

*Figure 2 continued on next page*

*Figure 2 continued*

MDA-MB-453 cells increased U/G MM repair-induced mutation. Western blot (left panel) shows NEIL2-HA overexpression level in MDA-MB-453 cells. Lamin B1 serves as a loading control. Error bar represents s.d., n = 3. $^*$P < 0.05 by two-tailed unpaired Student's *t* test.

The online version of this article includes the following source data, source code and figure supplement(s) for figure 2:

**Source code 1.** Python script for generating the volcano plot in *Figure 2A*.
**Source data 1.** Original data for DNA repair enzymes screening.
**Figure supplement 1.** TREX1 knockdown does not affect repair-induced mutagenesis.
**Figure supplement 2.** Repair-induced mutation rate is positively correlated with NEIL2 expression in breast cancer cells.

(*Hazra et al., 2002*), NEIL2 exhibited robust glycosylase/lyase activity on hydroxyl-U-containing single-stranded oligonucleotide (ssDNA-OHU), but only trace activity on double-stranded oligonucleotide (dsDNA-OHU/G) (*Figure 4—figure supplement 1B*). Furthermore, NEIL2 was inactive on both ssDNA-U and dsDNA-U/G (*Figure 4—figure supplement 1B*). However, the NEIL2 lyase can cleave the AP sites (*Figure 4—figure supplement 1C*) generated by UDG from U-containing oligonucleotides (*Figure 4—figure supplement 1D*).

APE1 is the conventional BER AP lyase. Therefore, we determined if NEIL2 collaborates or competes with APE1 for AP sites. APE1 and NEIL2 generate respectively 3'OH and 3'P-terminated fragments (*Figure 4A* and *Figure 4—figure supplement 2*), which migrate differently on Urea-PAGE (*Schomacher et al., 2016*). In a reaction with both NEIL2 and APE1, the NEIL2 product prevailed over the APE1 product (lane 5 of *Figure 4B*). Here we used amounts of the proteins just sufficient to completely digest the UDG-generated AP substrates from ss-U oligonucleotides (*Figure 4—figure supplement 3A,B*, lane 7).

In contrast to its activity on ssDNA, NEIL2 generated an unexplained double-band product from the dsDNA-U/G oligonucleotides in the concentration gradient assay (*Figure 4C*). Double band NEIL2 products have been previously observed (*Hazra et al., 2002*) and in our case were related to the length and composition of the substrate (35 nt, *Figure 4C* and 51nt, *Figure 4—figure supplement 3C*). Though unexplained, these distinct banding patterns provided a convenient way for distinguishing the NEIL2 and APE1 products. To determine whether NEIL2 can compete with APE1 at AP sites generated from dsDNA-U/G oligonucleotides, we gradually increased the concentration of NEIL2 in reactions that contained constant amounts of UDG and APE1. When APE1 was somewhat limiting (0.005U, Low APE1 panel, lanes 4–8, *Figure 4D*), the amount of NEIL2 products increased with increasing levels of NEIL2. Notably, the same results were found with excess APE1 (0.02U, High APE1 panel, lanes 9–13, *Figure 4D*). These results strongly suggest that NEIL2 supplants APE1 at AP sites. If NEIL2 was merely inhibiting APE1 activity, then the NEIL2 products would likely prevail only at the lower APE1 concentration.

## The NEIL2 product is a poor polβ substrate

As the Polβ reaction is a rate-limiting step during BER (*Srivastava et al., 1998*), we compared the NEIL2 and APE1 products as substrates for this reaction using His-tagged Polβ purified from *E. coli* (*Figure 4—figure supplement 4A*). The NEIL2 product retains a 3'P that can be removed by the phosphatase activity of PNKP (*Wiederhold et al., 2004*). Although Polβ showed robust activity on the APE1 product (*Figure 4—figure supplement 4B*), it was less active on the NEIL2 product, which depended on PNKP (*Figure 4E* and illustration in *Figure 4—figure supplement 2*). As expected from the usurpation of AP sites by NEIL2 (*Figure 4B,D*), Polβ incorporation was attenuated by NEIL2 (*Figure 4F*).

To confirm that our preparation of PNKP contained robust 3'-phosphatase activity, we prepared an oligonucleotide substrate that would yield a 3'-$^{32}$P-terminated product after completion of the coupled glycosylase/scission reactions of bifunctional glycosylases (*Wiederhold et al., 2004*) (*Figure 4—figure supplement 4C,D*). As a positive control, the cleavage product generated from this substrate by the *E. coli* bifunctional glycosylase, Fpg, yielded a 3'-$^{32}$P-terminated product that was equally susceptible to the phosphatase activity of either PNKP or T4 polynucleotide kinase (*Figure 4—figure supplement 4E*). However, in contrast to the complete cleavage of this substrate by

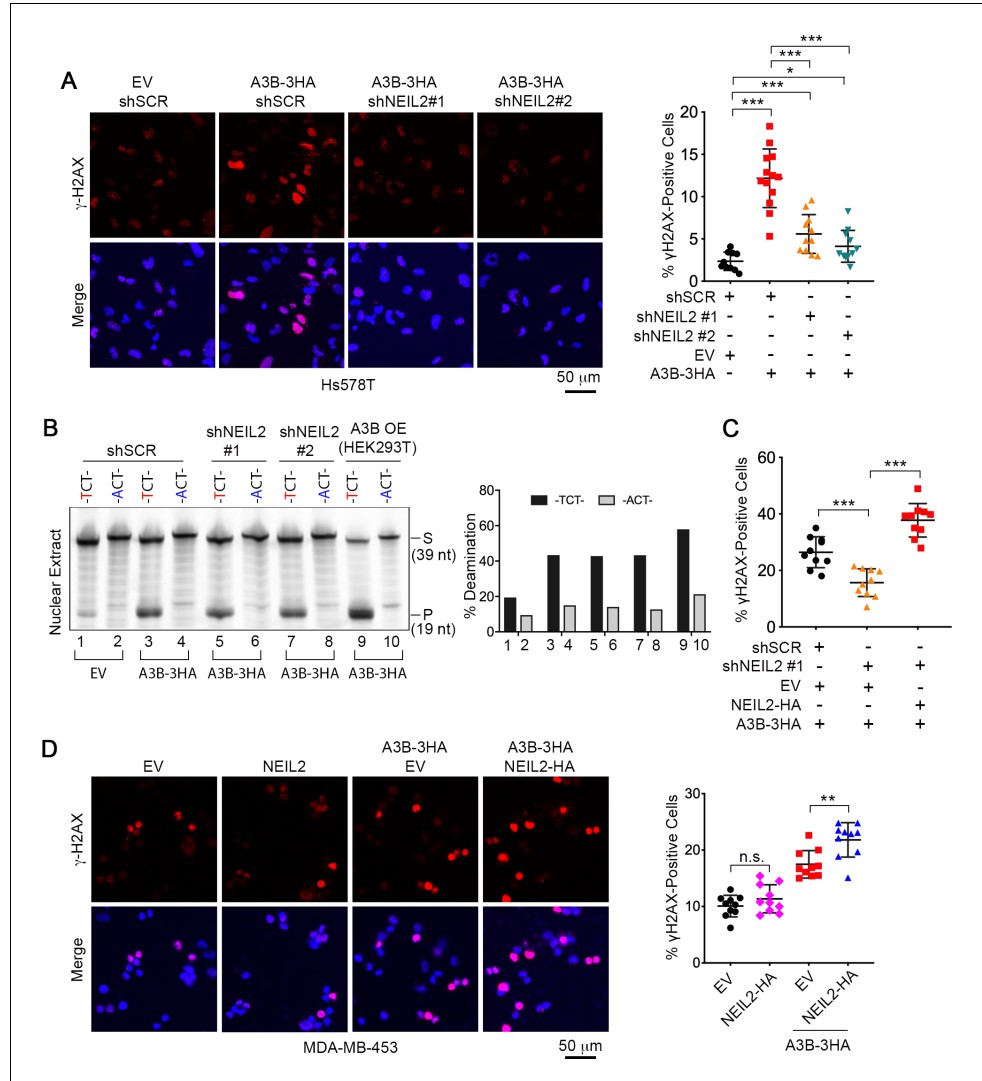

**Figure 3.** NEIL2 participates in A3B-mediated genomic DNA damage. (**A**) Immunostaining of γH2AX foci in NEIL2-stable-knockdown Hs578T cell lines (shNEIL2#1 and shNEIL2#2) transfected with A3B-3HA. NEIL2 knockdown decreases the A3B-mediated γH2AX foci. EV, empty vector; shSCR, scramble shRNA. Scale bar, 50 μm. Right panel: Percentage of γH2AX foci, showing mean ± s.d., in at least 10 randomly selected microscopic fields in two replicate experiments for each condition. ***P < 0.001 by two-tailed unpaired Student's *t* test. (**B**) In vitro deamination assay of nuclear extracts from NEIL2-stable-knockdown Hs578T cell lines with or without A3B-3HA expression. The substrate was a fluorescein-labeled single-stranded oligonucleotide (39 nt) containing -TCT- or -ACT- (negative control). Nuclear extract from HEK293T expressing A3B-3HA (A3B OE) was used as a positive control. NEIL2 knockdown does not affect A3B deaminase activity. Right panel: Quantifications of the cleaved products relative to total DNA loaded onto gel. S, substrate; P, product. (**C**) Quantification of the percentage of cells with γH2AX foci in NEIL2-stable-knockdown Hs578T cell line (shNEIL2#1) in the absence or presence of a NEIL2 expression vector pcDNA3.1(+)-NEIL2-3'HA. NEIL2 restoration increases A3B-triggered γH2AX foci. Data are represented as mean ± s.d. (n = 10 randomly selected microscopic fields in two replicate experiments). ***P < 0.001 by two-tailed unpaired Student's *t* test. The corresponding images of γH2AX foci are shown in *Figure 3—figure supplement 1C*. (**D**) Immunostaining of γH2AX foci in MDA-MB-453 cells overexpressing A3B-3HA and NEIL2-HA. Percentage of cells with γH2AX foci is shown in the right panel. EV, empty vector. Scale bar, 50 μm. Data are represented as mean ± s.d. (n = 10 randomly selected microscopic fields in two replicate experiments). **P < 0.01; n.s., no significant difference by two-tailed unpaired Student's *t* test.

The online version of this article includes the following figure supplement(s) for figure 3:

**Figure supplement 1.** NEIL2 is required for A3B-mediated genomic DNA damage.

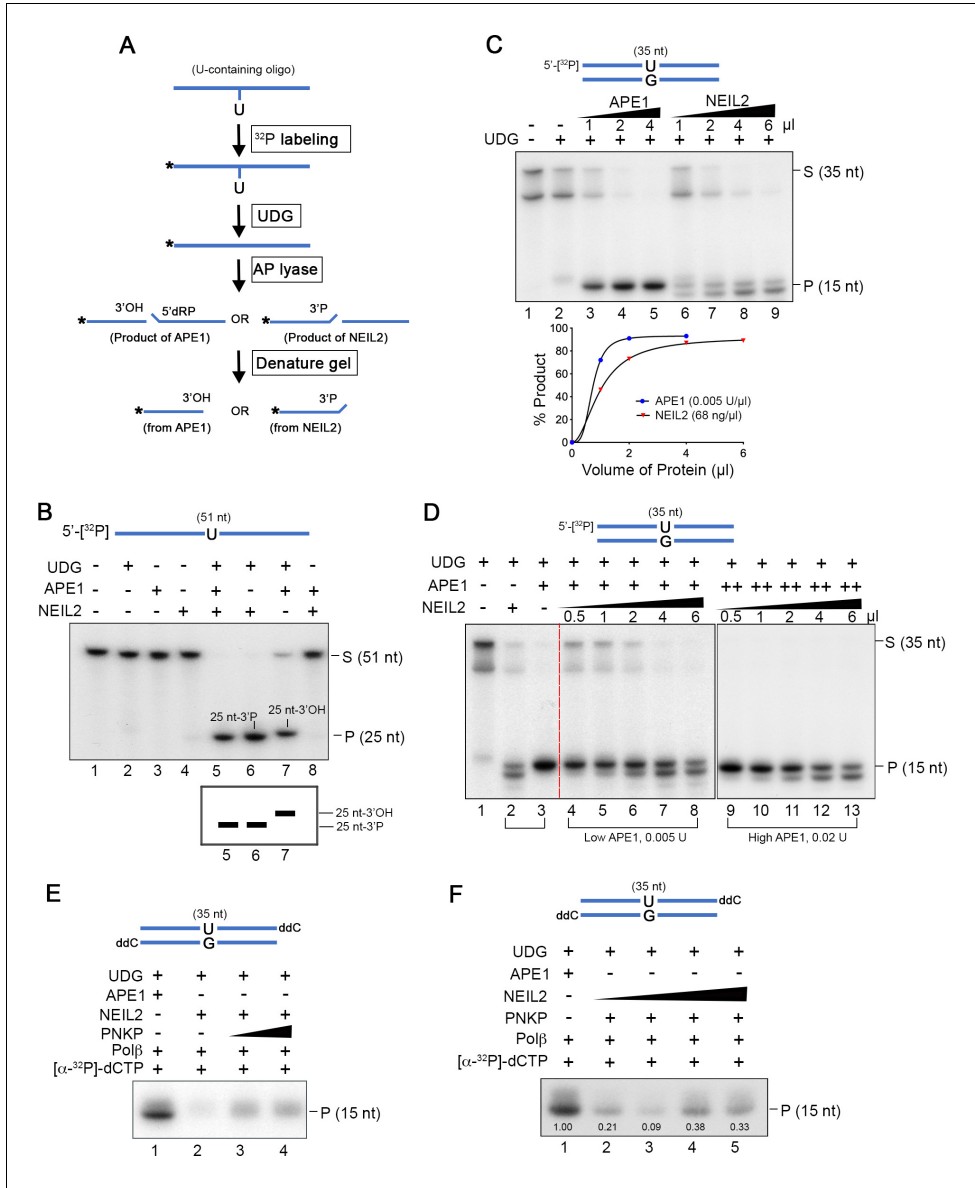

**Figure 4.** NEIL2 outcompetes APE1 at AP sites and the NEIL2 product is poor Polβ substrate. (**A**) Diagram of AP lyase assay. (**B**) NEIL2 outcompetes APE1 on AP sites generated from 5'-[$^{32}$P]-U-containing ssDNA (51 nt) by UDG. Amounts of NEIL2-His$_6$ or APE1 are just sufficient to completely cleave the AP site (lane 7 of *Figure 4—figure supplement 3A,B*). The NEIL2 product retains a 3'P and migrates faster than 3'OH-terminated APE1 product. When both NEIL2 and APE1 are present, only the NEIL2 product is generated (lane 5). S, substrate; P, product. (**C**) Concentration gradient and product accumulation curves of APE1 and NEIL2 on 5'-[$^{32}$P]-U-containing dsDNA (35 nt) in the presence of UDG. The volumes of NEIL2 (68 ng/μl) and APE1 (0.005 U/μl) used are listed in the figure. (**D**) NEIL2 competes with APE1 on 5'-[$^{32}$P]-U-containing dsDNA (35 nt) in the presence of UDG. The reactions contained either 0.005 U (Low APE1) or 0.02 U APE1 (High APE1) and increasing amounts of NEIL2-His$_6$. Lanes 2 and 3 contain respectively 4 μl NEIL2 (68 ng/μl) and 0.02 U APE1. The APE1 cleavage pattern was converted to the NEIL2 pattern with increasing amounts of NEIL2. S, substrate; P, product. (**E**) Incorporation of [α-$^{32}$P]-dCTP by Polβ for products generated by APE1, NEIL2, and NEIL2 and PNKP on di-deoxynucleotide (ddC)-modified oligonucleotide in the presence of UDG. P, product. (**F**) Incorporation of [α-$^{32}$P]-dCTP by Polβ in the presence of UDG and APE1 as a function of NEIL2 (68 ng/μl, 0, 1, 2, 5, 10 μl) and constant PNKP (127 ng/μl, 2 μl). As the NEIL2 product increasingly dominates the reaction, Polβ incorporation decreases. The number under each band gives the intensity relative to that in the first lane. P, product.

The online version of this article includes the following figure supplement(s) for figure 4:

**Figure supplement 1.** Purification and activity of NEIL2.

*Figure 4 continued on next page*

**Figure supplement 2.** Reaction mechanisms for the APE1 and NEIL2-PNKP pathways.
**Figure supplement 3.** Concentration gradients of APE1 and NEIL2 on AP sites.
**Figure supplement 4.** Purification and activity assays of PNKP and Polβ.

Fpg and to NEIL2's complete cleavage of a 5'-terminally labeled substrate (*Figure 4B–D* and *Figure 4—figure supplement 3B,C*), the internally-labeled substrate was resistant to complete cleavage by NEIL2 (*Figure 4—figure supplement 4E*, lanes 4 and 5). Perhaps these results reflect an isotope effect on NEIL2 activity.

## Exo1 generates single-stranded substrates vulnerable to A3B and DNA damage from NEIL2 products

We previously showed that hijacking of BER by mismatch repair (MMR) provided the wherewithal for Exo1 to generate single-stranded DNA for A3B (*Chen et al., 2014*). However, while NEIL2 has no glycosylase activity toward U/G (*Figure 4—figure supplement 1B*), it can interrupt normal BER by displacing APE1 at AP sites. As Exo1 can be recruited to endonucleolytic nicks generated by various means (*Wang et al., 2018*), we determined whether Exo1 was involved in NEIL2-mediated A3B-induced mutations or γH2AX foci. Exo1 siRNA knockdown (*Figure 5A*) reduced production of γH2AX foci in Hs578T cells but had no effect in the NEIL2-stable-knockdown cell lines (*Figure 5B,C*), indicating that Exo1 activity occurs downstream of NEIL2. In addition, U/G repair-induced mutations were also decreased in Exo1-knockdown Hs578T cells (*Figure 5D*). Thus, knockdown of Exo1 recapitulated the effects of NEIL2 knockdown. Furthermore, the nicked AP products generated by APE1 or NEIL2 were equally good substrates for Exo1 despite the different 5' termini of the nicked DNA (*Figure 5E*, 5'dRP for APE1 and 5'P for NEIL2). These results indicate that NEIL2 can divert a BER intermediate to an Exo1-generated single-stranded DNA that is susceptible to A3B deaminase activity, DSBs, or other DNA damage (*Figure 5F*).

## Discussion

A major unresolved issue in cancer biology is why some cancers become vulnerable to the mutagenic effect of A3 single-stranded deaminases (*Alexandrov et al., 2013b*; *Cescon et al., 2015*; *Helleday et al., 2014*; *Petljak et al., 2019*; *Roberts and Gordenin, 2014*), and even more intriguing, that it can be sporadic over clonal lineages of the same tumor (*Petljak et al., 2019*). Although it may be unlikely that a single mechanism would explain every instance of A3-mediated mutations in cancers, we report here that elevated expression of the bifunctional glycosylase, NEIL2 (*Figure 2A*), sensitizes Hs578T breast cancer cells to two A3B-mediated effects, repair-induced mutations (*Figure 2B–F*) and DNA damage revealed by γH2AX foci (*Burns et al., 2013a*; *Landry et al., 2011*) (*Figure 3*). NEIL2 depletion mitigates both effects, but they are induced by its overexpression. In vitro BER experiments using purified components show that NEIL2, though unable to process U/G mismatches, subverts the BER process that repairs these lesions by usurping the normal BER endonuclease, APE1, at AP sites (the first BER product) (*Figure 4*). Although we have not determined why the NEIL2 scission product is a poor substrate for Polβ, the important issue is that it is vulnerable to the 5'−3' exonuclease Exo1 (*Figure 5E*), which generates single-stranded DNA susceptible to A3 deaminases. In vivo experiments show that Exo1 and NEIL2 double-knockdown mimics the effect of NEIL2 depletion, indicating that NEIL2 acts upstream of Exo1 (*Figure 5A–C*).

Therefore, the most parsimonious explanation of our results is that NEIL2 diverts BER to Exo1-generation of single-stranded DNA that would be vulnerable to A3B deaminase (*Figure 5F*). It is also important to stress that the eventual outcome at a given AP site is not likely to be only a function of the relative intracellular concentrations of each protein. APE1 does have other binding partners (*Bazlekowa-Karaban et al., 2019*; *Madlener et al., 2013*; *Tell et al., 2009*; *Thakur et al., 2014*) and we presume this is likely also to be true of NEIL2 (*Das et al., 2007*). In addition, our results shows that NEIL2, even at a concentration of APE1 that is in four-fold excess of the amount needed to completely digest an AP site, can enzymatically outcompete APE1 (*Figure 4D*). The

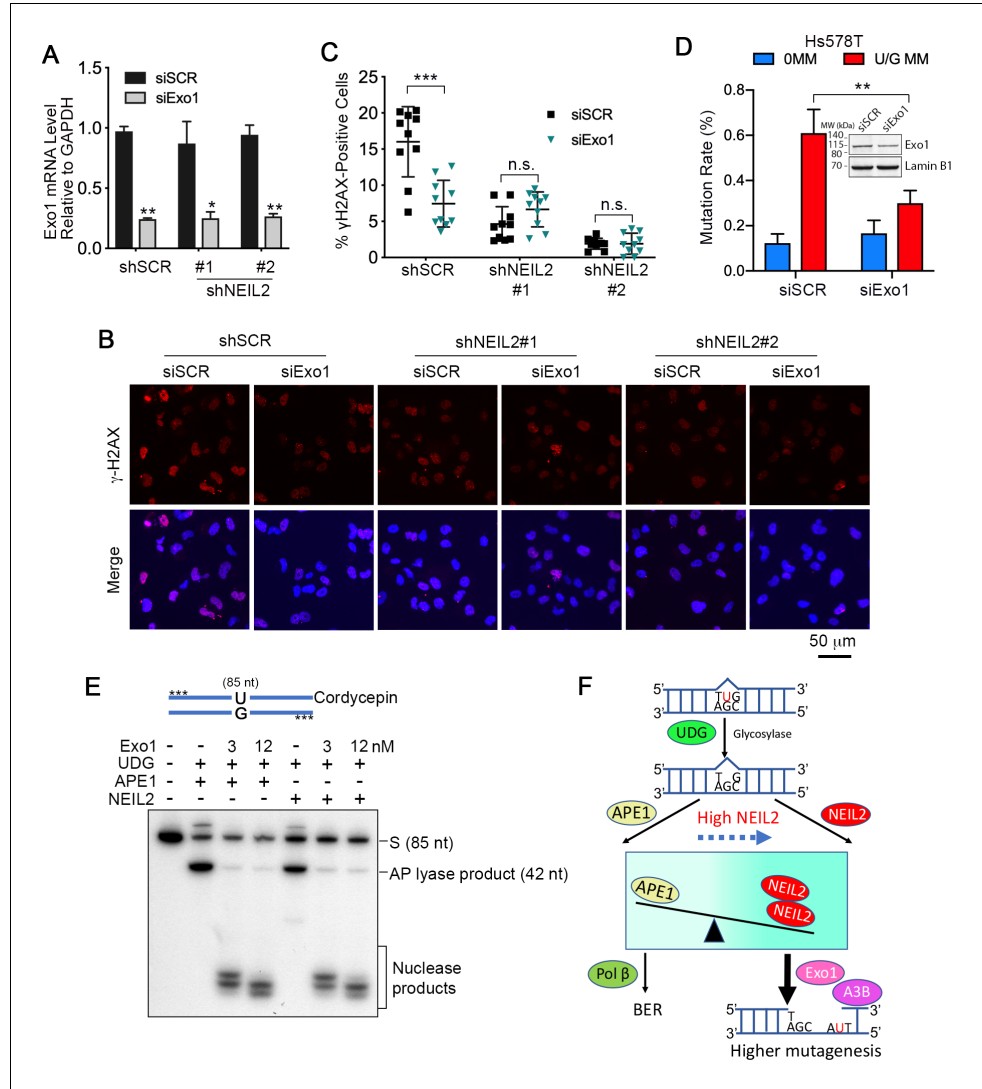

**Figure 5.** Exo1 generates single-stranded substrates vulnerable to A3B and DNA damage from NEIL2 products. (**A**) qRT-PCR analysis of siRNA knockdown of Exo1 in NEIL2-stable-knockdown Hs578T cell lines (shNEIL2#1 and shNEIL2#2). shNEL2#1 targets NEIL2 3'UTR; shNEIL2#2 targets NEIL2 ORF; shSCR, scramble shRNA; siSCR, scramble siRNA. Error bars represent s.d., n = 3. *P < 0.05; **P < 0.01 by two-tailed unpaired Student's *t* test. (**B**) Immunostaining of γH2AX foci in Hs578T cells depleted of Exo1 by siRNA in NEIL2-stable-knockdown Hs578T cell lines. A3B-3HA was expressed in these cell lines for 48 h before immunostaining. Scale bar, 50 μm. (**C**) Quantification of the percentage of cells with γH2AX foci from (**B**). Data are represented as mean ± s.d. (n=10 randomly selected microscopic fields). ***P < 0.001; n.s., no significant difference by two-tailed unpaired Student's *t* test. (**D**) Effect of Exo1 depletion on U/G MM repair-induced mutation rate in Hs578T cells. Insert, western blot analysis of Exo1 knockdown efficiency by siExo1 (20 nM siRNA). Lamin B1 serves as a loading control. 0 MM, no mismatch; U/G MM, U/G mismatch. **P < 0.01 by two-tailed unpaired Student's *t* test. (**E**) APE1 and NEIL2 scission products serve equally well as Exo1 substrates. 5'-phosphorothioate-modified (denoted by asterisk) U-containing oligo (85 nt) was 3'-labeled with cordycepin and used as the substrate for UDG, APE1 or NEIL2, and Exo1(human Exo1 protein). Nuclease products are bracketed. (**F**) Model depicting NEIL2-diversion of BER to Exo1-mediated resection to generate single-stranded A3B substrate.

inherent reductive power of a biochemical experiment using purified components is its ability to reveal the prevailing baseline conditions of a process (*Figure 5F*).

Several papers reported the presence of single nucleotide polymorphisms (SNPs) in the promoter region of NEIL2 that affected its expression (*Benítez-Buelga et al., 2017*; *Kinslow et al., 2008*; *Kinslow et al., 2010*). We sequenced the 700 bp NEIL2 promoter region of the four cell lines that

we used, and only the non-mutagenic, low NEIL2-exspressing MDA-MB-453 contained a SNP (rs804271) that had been previously correlated with elevated NEIL2 expression and DNA damage in a BRCA1/2 background (*Benítez-Buelga et al., 2017*). Thus, additional factors contribute to the regulation of NEIL2 in these cells. We found a positive correlation between U/G repair-induced mutations and relative NEIL2 expression (*Figure 2—figure supplement 2A,B*), but no relationship to endogenous A3B expression (*Figure 1B*), in four breast cancer cell lines. Interestingly, it was recently reported that clonal lineages of MDA-MB-453 can undergo sporadic episodes of A3-mediated mutations (*Petljak et al., 2019*). Thus, it seems that our ATCC isolate of this line (HTB-131) was in a low mutation phase. However, both high and low mutagenic clonal lines from the same lineage of MDA-MB-453 are available (*Petljak et al., 2019*) and would seem to provide the ideal experimental material to investigate by the approaches and methods we report in this paper. Given the heterogeneity of tumor samples, cell lines derived from the breast cancer tissues should also be useful sources of experimental material for investigating the relationship between NEIL2 expression and the mutational processes in tumors.

# Materials and methods

## Key resources table

| Reagent type (species) or resource | Designation | Source or reference | Identifiers | Additional information |
|---|---|---|---|---|
| Cell line (*Homo-sapiens*) | Hs578T, breast cancer | ATCC | HTB-126, RRID: CVCL_0332 | Authenticated by ATCC |
| Cell line (*Homo-sapiens*) | MDA-MB-453, breast cancer | ATCC | HTB-131, RRID: CVCL_0418 | Authenticated by ATCC |
| Cell line (*Homo-sapiens*) | MCF7, breast cancer | ATCC | HTB-22, RRID: CVCL_0031 | Authenticated by ATCC |
| Cell line (*Homo-sapiens*) | HCC1569, breast cancer | ATCC | CRL-2330, RRID: CVCL_1255 | Authenticated by ATCC |
| Cell line (*Homo-sapiens*) | LentiX-293T | Clontech | 632180 | Mycoplasma free |
| Strain, strain background (*Escherichia coli*) | MBM7070 (*lacZ*$^{uag\_amber}$) | Gift from Dr Michael Seidman (NIH) | | Electroporation |
| Strain, strain background (*Escherichia coli*) | Rosetta (DE3) | Millipore | 70954–4 | Protein expression |
| Antibody | anti-HA (Rabbit polyclonal) | Sigma-Aldrich | H6908, RRID: AB_260070 | WB (1:1,000) |
| Antibody | anti-Exo1 (Rabbit polyclonal) | Proteintech | 16253–1-AP, RRID: AB_2278140 | WB (1:1000) |
| Antibody | anti-Lamin B1 (Rabbit polyclonal) | abcam | ab16048, RRID: AB_10107828 | WB (1:2,000) |
| Antibody | anti-Rabbit IgG-Peroxidase | Sigma-Aldrich | A0545, RRID: AB_257896 | WB (1:10,000) |
| Antibody | anti-Mouse IgG-Peroxidase | Sigma-Aldrich | A4416, RRID: AB_258167 | WB (1:5,000) |
| Antibody | anti-γH2AX | Cell Signaling | 2577, RRID: AB_2118010 | IF (1:800) |
| Antibody | Alexa Fluor 568 anti-Rabbit IgG Secondary Antibody | Invitrogen | A-11036, RRID: AB_143011 | IF (1:500) |

*Continued on next page*

*Continued*

| Reagent type (species) or resource | Designation | Source or reference | Identifiers | Additional information |
|---|---|---|---|---|
| Sequenced-based reagent | siSCR (negative control) | Dharmacon | | UAGCGACUAAACACAUCAA |
| Sequenced-based reagent | siA3B | Dharmacon | J-017322-08-0005 | A3B knockdown |
| Sequenced-based reagent | siA3C | Dharmacon *Muckenfuss et al., 2006* | | AAGCCAACGAUCGGAACGAAA |
| Sequenced-based reagent | siNEIL2 | Dharmacon | | GCGAGGAUGAUUCUGAGUA |
| Sequenced-based reagent | siTREX1 | Dharmacon | | ACAAUGGUGACCGCUACGA |
| Sequenced-based reagent | siExo1 | Dharmacon | | CAAGCCUAUUCUCGUAUUU |
| Commercial assay or kit | A3A_TaqMan | ThermoFisher Scientific | Hs02572821_s1 | |
| Commercial assay or kit | A3B_TaqMan | ThermoFisher Scientific | Hs00358981_m1 | |
| Commercial assay or kit | A3C_TaqMan | ThermoFisher Scientific | Hs00819353_m1 | |
| Commercial assay or kit | A3H_TaqMan | ThermoFisher Scientific | Hs00419665_m1 | |
| Commercial assay or kit | NEIL2_TaqMan | ThermoFisher Scientific | Hs00979610_g1 | |
| Commercial assay or kit | Exo1_TaqMan | ThermoFisher Scientific | Hs01116190_m1 | |
| Commercial assay or kit | TREX1_TaqMan | ThermoFisher Scientific | Hs03989617_s1 | |
| Commercial assay or kit | GAPDH_TaqMan | ThermoFisher Scientific | Hs02786624_g1 | |
| Commercial assay or kit | TBP_TaqMan | ThermoFisher Scientific | Hs00427620_m1 | |
| Recombinant DNA reagent | pLKO.1 | addgene | RRID: Addgene_10878 | shRNA construction |
| Recombinant DNA reagent | pMDLg/pRRE | addgene | RRID: Addgene_12251 | lentivirus packaging |
| Recombinant DNA reagent | pRSV-Rev | addgene | RRID: Addgene_12253 | lentivirus packaging |
| Recombinant DNA reagent | pMD2.G | addgene | RRID: Addgene_12259 | lentivirus packaging |
| Transfected construct (*Homo-sapiens*) | phAPOBEC3B-HA (A3B-3HA) | NIH AIDS Reagent Program | 11090 | A3B overexpression |
| Transfected construct (*Homo-sapiens*) | pPM-NEIL2-3'HA | abm | PV028075 | NEIL2 rescue in mutation rate assay |
| Transfected construct (*Homo-sapiens*) | pcDNA3.1(+)-NEIL2-3'HA | this paper | | NEIL2 rescue in γH2AX assay |
| Transfected construct (*Homo-sapiens*) | pET22b-NEIL2 | Gift from Dr Tapas Hazra (U of Texas) | | NEIL2 expression and purification from *E. coli* |

*Continued on next page*

*Continued*

| Reagent type (species) or resource | Designation | Source or reference | Identifiers | Additional information |
|---|---|---|---|---|
| Transfected construct (*Homo-sapiens*) | pET28a(+)-PNKP | this paper | | For PNKP expression and purification |
| Transfected construct (*Homo-sapiens*) | pET28a(+)-Polβ | this paper | | For Polβ expression and purification |

## Cell culture and transfection

Hs578T cells were maintained in DMEM (Gibco, 11965084) with 10% FBS (Gibco, 10082147) and 0.01 mg/ml Insulin (Sigma-Aldrich, I0516) at 37°C in 5% $CO_2$. MCF7 cells were maintained in MEM (Gibco, 11095080) with 10% FBS at 37°C in 5% $CO_2$. HCC1569 cells were maintained in DMEM with 10% FBS at 37°C in 5% $CO_2$. MDA-MB-453 cells were maintained in Leibovitz L-15 media (Gibco, 11415064) supplemented with 10% FBS at 37°C without $CO_2$. LentiX-293T and HEK293T cells (a gift from Dr Roland Owens, NIH) were maintained in DMEM with 10% FBS at 37°C in 5% $CO_2$. Lipofect-amine3000 (Thermofisher Scientific, L3000008) and Mirus TransIT-BrCa transfection reagent (Mirus, MIR 5504) were used for plasmid transfection in breast cancer cells. FuGENE6 (Promega, E2691) was used to transfect plasmids to LentiX-293T and HEK293T cells following the manufacturers' protocol.

## Mismatch plasmid construction

Mismatch plasmids were constructed as previously described (*Chen et al., 2014*). For preparation of the gapped vector, we digested 60 µg pSP189-SnA plasmid (*Chen et al., 2014*) with 120 U nicking endonuclease, Nt.BbvCI (NEB), at 37°C overnight, followed by hybridization with 1,200 pmol biotiny-lated-complementary oligo at 37°C for 1 hr. The hybridized product was captured by 3 mg Streptavi-din Magnetic Particles (Roche, 11641786001) with rotation at 37°C for 2 hr. The gapped vectors were then purified by phenol/chloroform/isoamyl alcohol (25:24:1, PCI) extraction and ethanol precipitation.

For 0 MM and U/G MM reconstitution, 540 pmol C- or U-containing oligonucleotides were annealed at 100-fold molar excess to 18 µg gapped vectors in annealing buffer (100 mM KOAc, 30 mM HEPES, pH 7.5) by incubation at 95°C for 3 min, 40°C for 4 hr, 35°C for 30 s, 30°C for 30 s, and then kept at 25°C. Annealed samples were ligated with T4 DNA ligase (NEB, M0202S) at room temperature (RT, 25°C) for 2 hr, followed by incubation with 0.2 U Klenow Fragment (3'exo⁻,NEB, M0212), 100 µM dNTP, and 1× NEBuffer 2 at 37°C for 10 min to repair any remaining gapped plas-mids, which are highly mutagenic (*Chen et al., 2014*). The mismatch plasmids were purified by PCI extraction and ethanol precipitation. The gapping and ligation efficiencies were monitored by KpnI digestion (one of the two sites is lost after gapping but restored after ligation, *Figure 1—figure supplement 1A*).

## Determination of repair-induced mutations

Control (0 MM) and mismatch (U/G MM)-containing plasmids (1.5 µg per well in 6-well plate) were transfected into breast cancer cells, extracted 48 hr later using the Wizard Plus SV Miniprep kit (Promega, A1460), and treated with DpnI (NEB, R0176) for 15 min at 37°C. Plasmids (5 ng) were electroporated into 20 µL MBM7070 competent cells. After transformation, the cells were recovered in S.O.C. medium (Invitrogen, 15544–034) for 1 hr at 37°C and plated on LB agar plates containing 100 µg/mL carbenicillin (Sigma-Aldrich, C1389), 1 mM IPTG (Invitrogen, 15529–019), and 0.03% (w/v) Bluo-Gal (Invitrogen, 15519–028). After incubation at 37°C overnight, the plates were stored at 4°C in the dark to allow color development. The percentage of white to total colonies (2,000–3,000) per sample was calculated as 'mutation rate'. For mutation fate and trinucleotide context analysis shown in *Figure 1D,E*, we sequenced the reporter region (*Figure 1A*) of the pSP189-SnA shuttle vector in all white colonies (ACGT, Inc) using primer R250 (TTTTTGTGATGCTCGTCAGG) (*Chen et al., 2014*). Mutations were tabulated from the alignments between these sequences and the reference sequence of the starting reporter region.

## DNA repair enzymes screening and analysis

Breast cancer cells were harvested and cDNA was generated as described in the section of 'RNA isolation and qRT-PCR'. The cDNA was quantified by quantitative real-time PCR (qRT-PCR) using RT$^2$ Profiler PCR array Human DNA Repair (Qiagen, PAHS-042Z). Raw Ct values were normalized to two housekeeping genes, GAPDH and RPLP0, to determine $\Delta$Ct values. MDA-MB-453 $\Delta$Ct values were subtracted from Hs578T $\Delta$Ct values to obtain $\Delta\Delta$Ct values. Log$_2$(fold change) (calculated from log$_2$($2^{-\Delta\Delta Ct}$)) was plotted as a function -log$_{10}$(p-value) to generate the volcano plot (*Figure 2A*). These values were calculated from $\Delta$Ct values (n = 4 replicates) using the ttest_ind function from the Python scipy stats package. Only log$_2$(fold change) values above 2 or below −2 and -log$_{10}$(p-value) > 2 were counted as significant. The python script for generating the volcano plot in *Figure 2A* has been uploaded as an additional data file.

## RNAi

As preliminary experiments showed no difference in the reduction of NEIL2 transcripts between 10, 20 or 50 nM siRNA transfected with Lipofectamine RNAiMAX (ThermoFisher Scientific, 13778030) using the manufacturer's protocol (*Figure 2—figure supplement 1B*), we routinely used 10 nM siRNA unless stated otherwise. Cells were harvested 72 hr post-transfection for Western Blot or qRT-PCR analysis. The sequence information for siSCR, siA3B, siA3C, siNEIL2, siTREX1, and siExo1 can be found in the key resources table.

## RNA isolation and qRT-PCR

Total RNA was extracted using the PureLink RNA Mini Kit (Invitrogen, 12183018A), and reverse transcribed using the SuperScript III First-Strand Synthesis System (Invitrogen, 18080–051) following the manufacturer's instruction. The qRT-PCR was performed on StepOnePlus Real-Time PCR System (Applied Biosystems) using TaqMan gene expression assays (listed in the key resources table) and TaqMan Fast Universal PCR Master Mix (Applied Biosystems, 4352042). The $2^{-\Delta\Delta Ct}$ method (*Livak and Schmittgen, 2001*) was used to analyze the data.

## Lentivirus packaging and generation of stable knockdown cell lines

We used the 3$^{rd}$ generation packaging system following the addgene pLKO.1-TRC Vector protocol. LentiX-293T cells were transfected with 4 μg pLKO.1 vector containing short hairpin RNA (shRNA) in a 10 cm dish plate along with three packaging plasmids: pMDLg/pRRE (4 μg), pRSV-Rev (2 μg), and pMD2.G (2 μg). Media were replaced with 6 mL fresh complete media 24 hr post-transfection. After another 48 hr, media were harvested and centrifuged at 1,500 ×g for 10 min to remove cells and debris. The viral particles-containing supernatant and 7.5 μg/μl polybrene (Millipore, A-003-E) were added to infect Hs578T cells cultured in 6-well plate (100 μl per well). Media were replaced 24 hr after infection, and fresh media with 2 μg/ml puromycin (Sigma-Aldrich, P8833) were added another 24 hr later to select transfected cells.

shSCR_F: CCGGGTGGACTCTTGAAAGTACTATCTCGAGATAGTACTTTCAAGAGTCCACTTTTTG
shSCR_R: AATTCAAAAAGTGGACTCTTGAAAGTACTATCTCGAGATAGTACTTTCAAGAGTCCAC
shNEIL2#1_F: CCGGTCAGAGGTGCCAGTAGTATAACTCGAGTTATACTACTGGCACCTCTGATTTTTG
shNEIL2#1_R: AATTCAAAAATCAGAGGTGCCAGTAGTATAACTCGAGTTATACTACTGGCACCTCTGA
shNEIL2#2_F: CCGGGGGGCAGCAGTAAGAAGCTACACTCGAGTGTAGCTTCTTACTGCTGCCCTTTTTG
shNEIL2#2 _R: AATTCAAAAAGGGCAGCAGTAAGAAGCTACACTCGAGTGTAGCTTCTTACTGCTGCCC
shNEL2#1 targets NEIL2 3'UTR, and shNEIL2#2 targets NEIL2 ORF. All oligonucleotides used in this paper were synthesized by Integrated DNA Technologies (IDT) unless otherwise stated.

## Expression plasmids

Human NEIL2, PNKP and Polβ cDNAs were amplified from Hs578T cDNA using Q5 High-Fidelity DNA Polymerase (NEB, M0491) with the indicated forward and reverse primers and cloned into the indicated vectors:

NEIL2

F-NEIL2-EcoRI: TTT<u>GAATTC</u>ATGCCAGAAGGGCCGTTGGTG

R-NEIL2-XhoI: AAA<u>CTCGAG</u>GGAGAACTGGCACTGCTCTGG

pcDNA3.1(+)−3'HA vector (with a HA tag inserted at the C-terminus, a gift from Dr. Zhengfan Jiang at Peking University).

PNKP

F-PNKP-EcoRI: AAA<u>GAATTC</u>ATGGGCGAGGTGGAGGCCC

R-PNKP-HindIII: AAA<u>AAGCTT</u>TCAGCCCTCGGAGAACTGGC

Polβ

F-Polβ-BamHI: TTT<u>GGATCC</u>ATGAGCAAACGGAAGGCGCCG

R-Polβ-HindIII: AAA<u>AAGCTT</u>TCATTCGCTCCGGTCCTTGGGTT

Both cDNAs were cloned onto pET28a(+) vector (His-tagged, kindly provided by Dr. Wei Yang at NIH).

The underlined nucleotides denote restriction sites.

## Western blotting

Cells were harvested and centrifuged at 2,000 $\times g$ for 2 min. For nuclear proteins, cells were lysed in RIPA buffer (25 mM Tris-HCl, pH 7.6, 150 mM NaCl, 1% NP-40, 1% Sodium deoxycholate, 0.1% SDS) supplemented with protease inhibitor cocktail (Roche, 04693159001), and centrifuged at 13,000 $\times g$ for 10 min at 4°C. For western blotting, the proteins were separated on NuPAGE gels (Novex) and transferred to PVDF membranes (Novex, LC2002). After blocking with 3% BSA (in TBST: 50 mM Tris-Cl, pH 7.5, 150 mM NaCl, and 0.06% Tween-20), membranes were probed with primary antibodies at 4°C overnight. The membranes were washed three times in TBST (with 0.06% Tween-20), incubated with anti-Rabbit or anti-Mouse IgG-Peroxidase at RT for 1.5 hr, then treated for 5 min with SuperSignal West Pico Chemiluminescent Substrate (ThermoFisher Scientific, 34580) and exposed to X-ray film.

## Immunofluorescence

Hs578T cells seeded in 6-well plates were transfected with 1.5 µg phAPOBEC3B-HA (A3B-3HA) plasmid, and 24 hr post-transfection, the cells were seeded into Lab-Tek II Chamber Slide System (Nunc, 177380). After another 24 hr, the cells were washed with pre-warmed PBS, and fixed in 4% Formaldehyde (ThermoFisher Scientific, 28906, diluted with PBS) at 37°C for 15 min. Cells were washed three times with PBS and then permeabilized in 0.1% Triton X-100 (Sigma-Aldrich, T8787) at RT for 10 min, followed by another wash with PBS. For immunostaining, cells were blocked with 1% BSA (in PBS) at RT for 30 min, and incubated with anti-γH2AX primary antibody overnight at 4°C. After washing with PBS three times, cells were blocked with 1% BSA (in PBS) at RT for 30 min and then incubated with Alexa Fluor 568 anti-Rabbit IgG secondary antibody at RT for 2 hr. Cells were washed with PBS three times, mounted with Prolong Diamond Antifade Mountant with DAPI (Life technologies, P36962) and cured overnight at RT. Slides were imaged using a Keyence Digital Microscope and images were analyzed with Fiji (ImageJ) software using identical acquisition parameters for all images.

## Effect of expressing NEIL2-HA in NEIL2-stable-knockdown Hs578T cells

NEIL2-stable-knockdown (shNEIL2#1) and the negative control scramble shRNA-stable Hs578T cell lines were transfected with 1 µg pcDNA3.1(+)-NEIL2-3'HA or empty vector on the same day of cell seeding in 6-well plates. After 16 hr, the cells were transfected with 1.5 µg phAPOBEC3B-HA. The cells were re-seeded into a Nunc Lab-Tek II Chamber Slide System and underwent the γH2AX immunofluorescence procedure 24 hr later as described above.

## Purification of recombinant proteins

This purification protocol was kindly provided by Dr. Tapas Hazra at University of Texas. Plasmid pET22b-NEIL2, pET28a(+)-PNKP, or pET28a(+)-Polβ was transformed into Rosetta DE3 competent cells and protein expression in 500 mL cultures was induced with 0.5 mM IPTG at 16℃ overnight. Cell pellets were collected by centrifugation, washed with PBS, and resuspended in 20 mL lysis buffer (25 mM Tris, pH 7.5, 500 mM NaCl, 1 mM EDTA, 10% glycerol, 10 mM β-mercaptoethanol, 5 mM imidazole, 0.25% Tween-20, and 1 mM PMSF). Suspensions were sonicated on ice (40% Amp, 30 s ON and 30 s OFF) for 5 min, followed by addition of Triton X-100 to a final concentration of 0.25%. The lysate was centrifuged at 4,000 rpm at 4℃ for 30 min. The supernatant was applied to pre-equilibrated Ni-NTH agarose resin (Qiagen, 30230) and rotated for 2 hr in cold room. The resin was washed three times with increasing concentrations of imidazole (10 mM, 20 mM, and 40 mM) in wash buffer (25 mM Tris, pH7.5, 500 mM NaCl, 1 mM EDTA, 10% glycerol, 10 mM β-mercaptoethanol, and 0.25% TritonX-100). Elution buffer (25 mM Tris, pH 7.5, 300 mM NaCl, 0.5 mM EDTA, 10% glycerol, 10 mM β-mercaptoethanol, 0.25% TritonX-100, and 400 mM imidazole, 800 μl) was added to the resin and rocked at 4℃ for 30 min. The resin was centrifuged at 600 rpm for 3 min at 4℃ and the supernatant was retained. Another 800 μl of elution buffer were added to the matrix for a second elution, and the eluents were combined and dialyzed against 1 L of storage buffer (25 mM Tris, pH 7.5, 300 mM NaCl, 0.1 mM EDTA, 50% glycerol, 0.25% TritonX-100, and 2 mM TCEP) in a D-Tube Dialyzer (Millipore, 71510–3) with constant stirring at 4℃. Purified proteins were quantified using Pierce BCA Protein Assay Kit - Reducing Agent Compatible kit (ThermoFisher Scientific, 23250), aliquoted, and stored at −20℃.

## DNA deaminase activity

Deamination assays were performed as previously described (*Byeon et al., 2016*; *Mitra et al., 2014*). For nuclei isolation, breast cancer cells were harvested, washed with 1× PBS. The cells were resuspended in hypotonic buffer (20 mM Tris, pH7.5, 0.1 mM EDTA, 2 mM MgCl$_2$) and incubated at RT for 2 min and then on ice for 10 min, followed by adding 1/10 vol of 10% NP-40 (ThermoFisher Scientific, 28324) and centrifugation at 1,000 ×g for 5 min at 4℃ to pellet the nuclei. Extracts of HEK293T cells expressing A3B-3HA or isolated breast cancer nuclei were prepared by lysis in M-PER mammalian protein extraction reagent (ThermoFisher Scientific, 78501), supplemented with protease inhibitor cocktail (Roche, 04693159001) and 100 mM NaCl (final concentration). After lysis, glycerol (final concentration, 10%, vol/vol) was added and the resulting mixture was centrifuged at 13,000 ×g for 10 min at 4℃. Total protein concentration was quantified by Pierce BCA Protein Assay Kit (ThermoFisher Scientific, 23225).

Whole cell or nuclear extracts were incubated with RNase A (ThermoFisher Scientific, EN 0531) at 37℃ for 15 min before adding 500 nM fluorescein-labeled oligonucleotides (39 nt, synthesized by the Midland Certified Reagent Company, Inc) and 2U UDG (NEB, M0280) in 10 mM Tris-HCl pH 8.0, 50 mM NaCl, 1 mM DTT, 1 mM EDTA. Reactions were incubated at 37℃ for up to 5 hr followed by treatment with 150 mM NaOH at 37℃ for 20 min. After heating at 95℃ for 3 min, samples were immediately chilled on ice and purified by PCI extraction and ethanol precipitation. Samples were heated for 3 min at 95℃ with an equal volume of 2× Novex TBE-Urea Sample Buffer (Invitrogen, LC6876), separated by 12% 8M Urea PAGE gels, and imaged using the Fujifilm FLA-5100 (Fujifilm Life Science).

39 nt -TCT-:5'-fluorescein-AATAATAATAATAATAAT<u>TCT</u>AATAATAATAATAATAAT-3'
39 nt -ACT-:5'-fluorescein-AATAATAATAATAATAAT<u>ACT</u>AATAATAATAATAATAAT-3'

## $^{32}$P labeling and oligonucleotides annealing

For 5'-end $^{32}$P labeling, 30 pmol of oligonucleotide were incubated with 1× T4 PNK Buffer (NEB, B0201S), 20 U T4 Polynucleotide Kinase (NEB, M0201S), and 35 pmol [γ-$^{32}$P]-ATP (PerkinElmer, 3000 Ci/mmol) for 30 min at 37℃ followed by 65℃ for 20 min. For 3'-end $^{32}$P labeling, 20 pmol oligonucleotides were incubated with 1× TdT buffer (NEB, B0315S), 0.25 mM CoCl$_2$ (NEB, B0252S), 40 U terminal transferase (NEB, M0315S), and 40 pmol cordycepin (PerkinElmer, 800 Ci/mmol) for 30 min at 37℃ followed by 70℃ for 10 min. Unincorporated nucleotides were removed using Illustra Probe-Quant G-50 Micro Columns (GE Healthcare, 28903408) following the manufacturer's instructions. $^{32}$P-labeled oligonucleotides were annealed to 2-fold molar excess of unlabeled complementary

oligonucleotides in IDT annealing buffer (30 mM HEPES, pH 7.5, and 100 mM KOAc) by heating at 94°C for 2 min followed by a slow cooling to RT.

## NEIL2 and APE1 glycosylase/lyase activity assay

NEIL2 activity was measured as previously described (*Mandal et al., 2012*). Briefly, 0.5 pmol [32]P-labeled oligonucleotide substrate (35 nt or 51 nt) was mixed on ice with 1 µl UDG (1 U/µl) (NEB, M0280S), and purified NEIL2 (68 ng/µl) or APE1 (NEB, M0282S, 0.005 U/µl), and 1× NEBuffer 4 (50 mM KOAc, 20 mM Tris-Acetate, pH 7.9, 10 mM Mg(OAc)$_2$, 1 mM DTT) in a total reaction volume of 10 µl. Reactions were incubated at 37°C for 30 min. After PCI extraction and ethanol precipitation, samples were treated with equal volume of 2× Novex TBE-Urea Sample Buffer (Invitrogen). Cleaved products were separated using 12% 8 M Urea PAGE, then exposed to X-ray film.

For competition assays between NEIL2 and APE1, cleaved products were separated using 20% 8M Urea PAGE. For double-stranded DNA (dsDNA), 1 pmol [32]P-labeled dsDNA (35 nt) was mixed on ice with UDG (1 U/µl, 1 µl), various concentrations of purified NEIL2 or APE1 (indicated in figures and legends), and 1× NEBuffer 4 in 20 µl reactions. After incubation at 37°C for 30 min, the reaction products were extracted by PCI followed by ethanol precipitation. The samples were treated with an equal volume of 2× Novex TBE-Urea Sample Buffer (Invitrogen), heated at 95°C for 3 min, and chilled on ice before subjecting to 20% 8 M Urea PAGE.

51-nt single strand U:
5′-GCTTAGCTTGGAATCGTATCATGTA(U)ACTCGTGTGCCGTGTAGACCGTGCC-3′
51-nt single strand hydroxyl-U:
5′-GCTTAGCTTGGAATCGTATCATGTA(OHU)ACTCGTGTGCCGTGTAGACCGTGCC-3′
51-nt bottom strand:
5′-GGCACGGTCTACACGGCACACGAGTGTACATGATACGATTCCAAGCTAAGC-3′
35-nt single strand U: 5′-GCCCTGCAGGTCGA(U)TCTAGAGGATCCCCGGGTAC-3′
35-nt bottom strand: 5′-GTACCCGGGGATCCTCTAGAGTCGACCTGCAGGG-3′

## Polβ incorporation assay

The substrate was the previously reported (*Beard et al., 2006*) 35 nt double-stranded oligonucleotide containing a U/G pair modified by addition of 3′-dideoxycytidine (3ddC) to block end incorporation. Incorporation reactions (20 µl) were assembled on ice and contained 1× NEBuffer 4 (50 mM KOAc, 20 mM Tris-Acetate, pH 7.9, 10 mM Mg(OAc)$_2$, 1 mM DTT), 25 nM 35 bp dsDNA, 1.25 U/µl UDG, $0.6 \times 10^6$ dpm/pmol [α-[32]P]-dCTP (1:10 diluted in 5 nM dCTP, 2 µl) and the indicated amounts of APE1 (0.01 U/µl, 1 µl), purified NEIL2 (68 ng/µl, 2.5 µl), PNKP (127 ng/µl, 2 µl) and Polβ (1.7 ng/µl, 2 µl). In the competitive incorporation assay (*Figure 4F*), the PNKP for each reaction was held constant and the amount of NEIL2 was increased as indicated in the figure legend. The reactions were initiated by adding the proteins, incubated at 37°C for 15 min, and then terminated with 100 mM EDTA, followed by PCI extraction and ethanol precipitation. The products were treated with an equal volume of 2× Novex TBE-Urea Sample Buffer (Invitrogen), separated using 12% Urea PAGE, and exposed to X-ray film.

Top_U_3ddC_35: GCCCTGCAGGTCGA(U)TCTAGAGGATCCCCGGGTA/ddC/
Bottom_G_3ddC_35: GTACCCGGGGATCCTCTAGAGTCGACCTGCAGGG/ddC/

## 3′ phosphatase activity of PNKP

A 26 nt oligonucleotide with U at the 5′ end was labeled with [32]P by T4 PNK (NEB) as described above. As illustrated in *Figure 4—figure supplement 4C*, the oligonucleotide was annealed to a 51 nt complementary oligonucleotide, and then annealed to a 25 nt oligonucleotide in annealing buffer (10 mM Tris, pH 7.5, 50 mM NaCl, 1 mM EDTA), followed by ligation with T4 DNA ligase (NEB) to form an internally-labeled duplex oligonucleotide (designated S, *Figure 4—figure supplement 4C*). The oligonucleotide (2 pmol) was treated with UDG (1 U, NEB) and Fpg (1U, NEB) or NEIL2 (272 ng) in a 10 µl reaction at 37°C for 30 min. The products were purified by PCI extraction and ethanol precipitation and treated with PNKP (25 mM HEPES-KOH, pH 7.6, 50 mM NaCl, 0.5 mM EDTA, 0.5 mM DTT, 100 ng/ml BSA, 5% Glycerol) or T4 PNK (70 mM Tris-HCl, 10 mM MgCl$_2$, 5 mM DTT, pH 6.0) at 37°C for 30 min. The reactions were stopped on ice for 10 min and then analyzed by 20% Urea PAGE as described above.

Top_25_left: GCTTAGCTTGGAATCGTATCATGTA
Top_26_U_right: UACTCGTGTGCCGTGTAGACCGTGCC
Bottom_51:GGCACGGTCTACACGGCACACGAGTGTACATGATACGATTCCAAGCTAAGC

## Exo1 nuclease assay

The dsDNA substrates for the resection assay were 85 nt oligonucleotides, which contained a 5'-phosphorothioate modification to block end resection by exonuclease. The 85 nt U-containing (U at position 43) oligonucleotides were 3'-labeled with [32]P-cordycepin and then annealed with the bottom strand before treatment with UDG and APE1 or NEIL2 as described above. After PCI purification and ethanol precipitation, 0.1 pmol dsDNA substrates were incubated with Exo1 (human Exo1 protein, kindly provided by Dr. Tanya Paull at the University of Texas at Austin) (*Myler et al., 2016*) resection assay in a 10 µl reaction as previously described (*Keijzers et al., 2015*). The reactions were carried out at 30℃ for 15 min and then purified by PCI extraction and ethanol precipitation. The products were mixed with an equal volume of 2× Novex TBE-Urea Sample Buffer (Invitrogen) and heated at 90℃ for 3 min. Samples were subjected to 20% Urea PAGE and exposed to X-ray film.

Top_U_5'phosphorothioate_85: 5'G*A*C*AGGATCCGGGCTAGCATCTTCATACGCCCTGCAGGTCGAUTCTAGAGGATCCCCGGGTACCTTCATACATAGTTGTCACTGG3'

Bottom_G_5'phosphorothioate_85: 5'C*C*A*GTGACAACTATGTATGAAGGTACCCGGGGATCCTCTAGAGTCGACCTGCAGGGCGTATGAAGATGCTAGCCCGGATCCTGTC3'

* denotes phosphorothioate

## Quantification and statistical analysis

Statistical details, including the statistical methods, n values, definition of significance, and definition of mean value and dispersion were indicated in the figure legends. Statistical analyses were carried out using GraphPad Prism 8.

## Acknowledgements

This work was supported by the Intramural Research Program of the National Institute of Diabetes and Digestive and Kidney Diseases (NIDDK) at the NIH. We thank Dr. Tapas Hazra at University of Texas for providing the pET22b-NEIL2 plasmid and detailed NEIL2 purification protocol, Dr. Tanya Paull at the University of Texas at Austin for providing the Exo1 protein, Dr. Wei Yang at the NIH for pET28a(+) plasmid, and Dr. Zhengfan Jiang at Peking University for pcDNA3.1(+)−3'HA plasmid. Thanks also to the NIDDK Advanced Light Microscopy and Image Analysis Core (ALMIAC) for the use of its resources.

## Additional information

### Funding

| Funder | Grant reference number | Author |
| --- | --- | --- |
| National Institute of Diabetes and Digestive and Kidney Diseases | Intramural Research Program | Anthony V Furano |

The funders had no role in study design, data collection and interpretation, or the decision to submit the work for publication.

### Author contributions

Birong Shen, Conceptualization, Data curation, Formal analysis, Supervision, Validation, Investigation, Visualization, Methodology, Writing - original draft, Project administration, Writing - review and editing; Joseph H Chapman, Conceptualization, Formal analysis, Validation, Investigation, Visualization, Methodology, Writing - review and editing; Michael F Custance, Conceptualization, Validation, Investigation, Visualization, Methodology; Gianna M Tricola, Data curation, Formal analysis, Validation, Investigation, Methodology, Writing - review and editing; Charles E Jones, Validation,

Investigation, Visualization, Methodology, Writing - review and editing; Anthony V Furano, Conceptualization, Resources, Data curation, Software, Formal analysis, Supervision, Funding acquisition, Validation, Investigation, Visualization, Methodology, Project administration, Writing - review and editing

## Author ORCIDs
Birong Shen (ID) https://orcid.org/0000-0002-1208-506X
Gianna M Tricola (ID) http://orcid.org/0000-0002-5965-8726
Anthony V Furano (ID) https://orcid.org/0000-0002-4489-6828

## Decision letter and Author response
Decision letter https://doi.org/10.7554/eLife.51605.sa1
Author response https://doi.org/10.7554/eLife.51605.sa2

## Additional files

### Supplementary files
• Transparent reporting form

### Data availability
All data generated or analysed during this study are included in the manuscript and supporting files.

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
