## [Decision Letter]

**Acceptance summary:**

In this manuscript, the authors investigated the mechanism underlying the variable frequency of mutagenesis induced by the APOBEC3B (A3B) deaminase in breast cancer. A known pathway for repair of U/G mismatches involves the removal of the uracil, creating an apurinic/apyrimidinic (AP) site that is further processed by the activity of AP endonuclease 1 (APE1) to generate nicks with a 3' OH group. Such nicks are recognized and repaired by DNA polymerase β activity followed by Ligase 1 or III/XRCC1 mediated ligation. The authors have discovered that in some cells, NEIL2 displaces APE1 to create a product with a 3' phosphate group that is unsuitable for processing by DNA polymerase β. This product triggers an alternative cascade of reactions involving exonuclease 1, forming single stranded DNA that is susceptible to A3B activity, thus enhancing mutagenesis and inducing double stranded DNA breakage. The enhancement of A3B activity by this interaction can help provide a mechanistic basis for the observation that in many cancers, A3B expression levels are not good predictors of A3B-induced mutagenesis.

**Decision letter after peer review:**

Thank you for submitting your article "Perturbation of base excision repair sensitizes breast cancer cells to APOBEC3 deaminase-mediated mutations" for consideration by *eLife*. Your article has been reviewed by three peer reviewers, one of whom is a member of our Board of Reviewing Editors, and the evaluation has been overseen by Jeffrey Settleman as the Senior Editor. The reviewers have opted to remain anonymous.

The reviewers have discussed the reviews with one another and the Reviewing Editor has drafted this decision to help you prepare a revised submission.

Summary:

The APOBEC3 deaminase-mediated mutation phenotype in breast cancer has been highlighted in many studies. In this manuscript, the authors investigated the mechanism underlying the variable frequency of mutagenesis induced by APOBEC3B (A3B) in breast cancer. A known pathway for repair of U/G mismatches involves the removal of the uracil, creating an apurinic/apyrimidinic (AP) site that is further processed by the activity of AP endonuclease 1 (APE1) to generate nicks with a 3'OH group. Such nicks are recognized and repaired via the activity DNA polymerase β followed by Ligase 1 or III/XRCC1 mediated ligation. The current study reports that in some cells, NEIL2 displaces APE1 to create a product with a 3' phosphate group that is unsuitable for processing by DNA polymerase β. This product triggers an alternative cascade of reactions involving exonuclease 1, forming single stranded DNA that is susceptible to A3B activity, thus enhancing mutagenesis and inducing double stranded DNA breakage. The enhancement of A3B activity by this interaction can help provide a mechanistic basis to the observation that in many cancers, A3B expression levels are not good predictors of A3B-induced mutagenesis.

The manuscript is written clearly and provides data that are generally supportive of the hypothesis, but several concerns need to be addressed. In addition, as currently written, certain methodological points are explained very succinctly, making the interpretation difficult. The manuscript will be improved by a clearer Introduction to the methodology and by including additional data, some of which includes cell-based assays, and statistical tests to address the concerns listed below.

Essential revisions:

1) The deaminase activity assay presented in Figure 1B comparing two different cell's nuclear extracts is showing fast saturation (not linear), authors might consider doing a time-kinetics to give a better representation and also add recombinant A3B protein as a control. Together with A3B mRNA expression, to connect with A3B function, authors should consider checking protein level in different breast cancer cell lines.

2) In all the deaminase activity assays HEK293T A3B O/E nuclear extract is showing much higher activity. Thus, authors should show NEIL2 and APE1 protein levels in those extracts to identify rate-limiting inhibitory components present in breast cancer cell lines and to support the model.

3) Figure 2F and 2G are either missing error bars or appear to be not statistically significant. Western blots need to be provided with all the deaminase assay of knockdown experiments and rescue experiments. If NEIL2 is substituting APE1 activity, then APE1KD should display higher mutator phenotype in these cells or overexpression of APE1 should decrease the mutation rate in NEIL2 overexpressing HS578T cells. Cell-based experiments are needed to support the competition between APE1 and NEIL2.

4) In Figure 3A, it seems that γH2AX staining increased in all cells upon the expression of A3B and that shNEIL2 diminished this effect but did not completely prevented it as written in the Results section. It would be good to modify the text to reflect this result. Figure 3D shows the results of a γH2AX assay in the non-mutagenic MDA-MB-453 that were subject to exogeneous expression of both A3B and NEIL2. The control shows a γH2AX assay in A3B expressing cells but the background of γH2AX expression in the parental cells is not shown. Because the experiments aim to measure A3B-mediated γH2AX foci, quantification of such foci in the parental cells is essential for interpreting the results. Additionally, the results from the γH2AX experiments in Figure 3A need to be further supported by immunoblotting for γH2AX and total H2AX. The same concern applies to the experiments in Figure 5B.

5) APE1 is known to be a poor enzyme for a single-stranded substrate as it needs a double strand to bind and cleave efficiently. Thus, that can explain APE1's reduced activity in Figure 4A. For double-stranded substrate APE1 activity seems much faster. Again, the kinetics seems little off as the substrate to product ratio is not consistent. It looks like double-stranded oligo is not a suitable substrate for NEIL2. APE1 is much faster, and a higher concentration of NEIL2 can compete for only 50% activity. Figure 4C and D is very complex to understand; in Figure 4C, it seems NEIL2 is bound to UDG substrate that inhibits APE1 activity. Thus, NEIL2 has slower turnover from the substrate. But Figure 4D demonstrates a higher amount of APE1 facilitates NEIL2 activity. In this regard, measuring dissociation constant from the substrate can be useful to dissect the issue of activity vs. enzyme turnover. Figure 4F aims to test whether processing by NEIL2 creates products that are poor substrates for DNA polymerase β. This is a key point in the paper and quantification is important. In the experiment reported in the current submission, NEIL2 levels vary along with PNKP levels, and interpretation is therefore difficult. It would be useful to perform the NEIL2 dose-response analysis in the presence of a constant level of PNKP.

6) A substrate using THF as an AP site substitute can be used as a control. Also using a substrate with 3'P rather than UDG can be a cleaner set-up to compare.

7) Although the authors stated that Exo1 activity is similar for both APE1 and NEIL2 products, Figure 5E shows different substrate to product ratios for two enzymes. Authors might consider adding a panel of APE1 and NEIL2 together in Exo1 activity assay in Figure 5E to understand if the presence of APE1 facilitates or inhibits Exo1 activity.

[Editors' note: further revisions were requested prior to acceptance, as described below.]

Thank you for resubmitting your work entitled "Perturbation of base excision repair sensitizes breast cancer cells to APOBEC3 deaminase-mediated mutations" for further consideration by *eLife*. Your revised article has been evaluated by Jeffrey Settleman (Senior Editor) and a Reviewing Editor.

The manuscript has been improved but there are some remaining issues that need to be addressed before acceptance, as outlined below:

The authors have addressed an important concern, providing better descriptions of the specialized methodology used in the paper. The authors have also removed some controversial data from the paper, resulting in a succinct, readable and focused report. However, some controls are still missing:

1).An experimental validation of the proposed competition between APE1 and NEIL2 (usurpation of APEI by NEIL2) is critical. A ChIP-seq or a chromatin binding assay can be used for this purpose. If validated, this point is potentially very important, as is emphasized in the Abstract.

2) Due to the absence of appropriate measurements of protein levels (for example, NEIL2 and APE1 in nuclear extracts that show variable activity in deaminase assays) it is not possible to rule out that the observations reported in the paper might be explained by alternative mechanisms. If appropriate antibodies are not available, these other mechanisms should at least be discussed.

3) The authors are correct to suggest that immunofluorescence is an acceptable measure of DNA damage in some literature. However, because of the high signal-to-noise ratio in the current submission, the paper would still benefit from another measurement of the extent of DNA damage based on a different method, for example, immunoblotting of γH2AX vs. total H2AX.

For the quantification of immunofluorescent signals in Figure 3 (A, right panel, C and D): what does the y-axis (% of γH2AX positive cells) refer to? Does each data point refer to a replicate experiment, or was the percentage of positive cells determined in each field in a single imaging experiment? How significant are the changes induced by the two shNEIL2 when compared to control levels?

---

## [Author Response]

Essential revisions:1) The deaminase activity assay presented in Figure 1B comparing two different cell's nuclear extracts is showing fast saturation (not linear), authors might consider doing a time-kinetics to give a better representation and also add recombinant A3B protein as a control. Together with A3B mRNA expression, to connect with A3B function, authors should consider checking protein level in different breast cancer cell lines.

We addressed the reviewer’s suggestions as follows: For Figure 1B in the original submission, we prepared new nuclear extracts from the two breast cancer cells, assayed them over a range of concentrations (0, 5, 15, 25, and 35 μg), and quantified the deamination percentage (Figure 1C in the revised manuscript). As recommended, we added A3B-expressing HEK293T cell extract as a control (Figure 1C in the revised manuscript). In addition, we performed a time course for the in vitro deamination assay using 25 μg of nuclear extract from the breast cancer cells over the following hourly time intervals: 0, 0.5, 1, 2, 4, and 5, using an A3B-expressing HEK293T cell extract as a control (Figure 1—figure supplement 1C in the revised manuscript).

We agree with the reviewer that relating the amount of deaminase protein is a more valid reflection of A3B activity than A3B mRNA. In an attempt to do so, we tested two commercial anti-A3B antibodies (abcam, ab184990; Proteintech, 14559-1-AP). While neither were sufficiently sensitive to recognize endogenous A3B, they did recognize exogenous A3B from an expression vector (Author response image 1). This probably explains why a number of A3B studies only report mRNA levels (e.g. Burns et al., 2013; Cescon et al., 2015; Law et al. Sci. Adv. 2016). Thus, we used the in vitro deaminase assay to measure A3B activity in the breast cancer cell lines (Figure 1C and Figure 3B in the revised manuscript).

**Author response image 1. respfig1:** Commercially available anti-A3B antibodies do not recognize endogenous A3B protein. (**A**) 100 μg whole cell lysate from MDA-MB-453 and Hs578T cells were subject to western blotting and detected by anti-A3B antibodies shown. Lamin B1 serves as a loading control. (**B**) 100 μg whole cell lysate from Hs578T cells overexpressing empty vector or A3B-3HA were subject to western blotting and detected by anti-A3B antibodies shown. Both antibodies recognize overexpressed A3B-3HA but not endogenous A3B. HEK293T overexpressing A3B-3HA lysate detected by anti-HA antibody was used as a positive control. LaminB1 serves as a loading control.

2) In all the deaminase activity assays HEK293T A3B O/E nuclear extract is showing much higher activity. Thus, authors should show NEIL2 and APE1 protein levels in those extracts to identify rate-limiting inhibitory components present in breast cancer cell lines and to support the model.

An essential element of our model is that the extent of U/G repair-induced mutations is not solely dependent on the level of A3B deaminase, but also, or even more so, on the availability of the A3B single-stranded substrate. We surmised this when we compared and found that the extent of U/G-repair induced mutations was not correlated with either the amount or the activity of A3B deaminase in nuclear extracts of breast cancer cells (Figure 1B,C in the revised manuscript). To measure the A3B deaminase activity for each cell line, we extracted their nuclear extracts and used the in vitro deamination assay as depicted in Author response image 2: equivalent amounts of A3B-preferred-TCT-containing single-stranded oligonucleotide was used as the substrate. As the deamination activity is only dependent on the amount of A3B in the nuclear extract, the cell extract of HEK293T cells overexpressing A3B (used as a positive control) would be expected to contain considerably higher activity than the nuclear extract of breast cancer cells (endogenous level, Figure 1C and Figure 1—figure supplement 1B,C in the revised manuscript).

**Author response image 2. respfig2:** Schematic of the in vitro deamination assay.

The deamination assays did not speak to the relative amounts of NEIL2 or APE1 in these extracts. After we obtained results in Figure 1B,C shown in the revised manuscript, we screened the breast cancer cell lines for their expression of DNA repair enzymes (Figure 2A in the revised manuscript) and found that repair-induced mutagenesis was correlated with elevated levels of NEIL2.

3) Figure 2F and 2G are either missing error bars or appear to be not statistically significant. Western blots need to be provided with all the deaminase assay of knockdown experiments and rescue experiments. If NEIL2 is substituting APE1 activity, then APE1KD should display higher mutator phenotype in these cells or overexpression of APE1 should decrease the mutation rate in NEIL2 overexpressing HS578T cells. Cell-based experiments are needed to support the competition between APE1 and NEIL2.

We thank the reviewer for pointing this out. We repeated the experiments and added error bars for the previous Figure 2F and 2G (Figure 2E,F in the revised manuscript). We also added western blots for the rescue experiments with NEIL2-HA overexpression (Figure 2E,F in the revised manuscript). However, for the NEIL2 knockdown, we were unable to detect the endogenous NEIL2 using any of the commercially available anti-NEIL2 antibodies that we had tested (abcam, ab180576 showed a non-specific band at ~50 kDa; abcam, ab124106 can only recognize the overexpressed NEIL2).

We also agree with the reviewer that it would be informative if one could design cell-based experiments to support the in vitro competition between APE1 and NEIL2 at AP sites. However, APE1, which aside from its endonuclease activity on AP sites, has numerous additional functions and interactions: including telomere maintenance: Madlener et al., 2013; as a reductive activator of many transcription factors, which is why it’s also referred as Ape1/Ref-1 (redox effector factor); and interacts with proteins and factors involved in energy metabolism and homeostasis: Thakur et al., 2014; Tell et al., 2009. Therefore, we could not design an in vivo experiment that would specifically address its endonuclease activity at AP sites versus that of NEIL2. In contrast, the in vitro experiments with purified components are unambiguous in this regard (Figure 4 in the revised manuscript).

4) In Figure 3A, it seems that γH2AX staining increased in all cells upon the expression of A3B and that shNEIL2 diminished this effect but did not completely prevented it as written in the Results section. It would be good to modify the text to reflect this result. In Figure 3D, panel shows the results of a γH2AX assay in the non-mutagenic MDA-MB-453 that were subject to exogeneous expression of both A3B and NEIL2. The control shows a γH2AX assay in A3B expressing cells but the background of γH2AX expression in the parental cells is not shown. Because the experiments aim to measure A3B-mediated γH2AX foci, quantification of such foci in the parental cells is essential for interpreting the results. Additionally, the results from the γH2AX experiments in Figure 3A need to be further supported by immunoblotting for γH2AX and total H2AX. The same concern applies to the experiments in Figure 5B.

We agree with the reviewer and rephrased our original text to “which was markedly decreased in NEIL2-depleted Hs578T cells”. For Figure 3D, we added two groups of cells and undertook the immunostaining experiment to show the background of γH2AX foci in the parental cells without A3B overexpression (Figure 3D in the revised manuscript). Regarding Figure 3A and Figure 5B, analysis of γH2AX foci by immunofluorescence is generally considered a sufficient and direct measurement of DNA double-strand breaks (e.g. Morel et al., 2017; Ladstatter et al, Cell. 2016; Kinner et al., Nucleic Acids Res. 2008).

5) APE1 is known to be a poor enzyme for a single-stranded substrate as it needs a double strand to bind and cleave efficiently. Thus, that can explain APE1's reduced activity in Figure 4A. For double-stranded substrate APE1 activity seems much faster. Again, the kinetics seems little off as the substrate to product ratio is not consistent. It looks like double-stranded oligo is not a suitable substrate for NEIL2. APE1 is much faster, and a higher concentration of NEIL2 can compete for only 50% activity. Figure 4C and D is very complex to understand; in Figure 4C, it seems NEIL2 is bound to UDG substrate that inhibits APE1 activity. Thus, NEIL2 has slower turnover from the substrate. But Figure 4D demonstrates a higher amount of APE1 facilitates NEIL2 activity. In this regard, measuring dissociation constant from the substrate can be useful to dissect the issue of activity vs. enzyme turnover. Figure 4F aims to test whether processing by NEIL2 creates products that are poor substrates for DNA polymerase β. This is a key point in the paper and quantification is important. In the experiment reported in the current submission, NEIL2 levels vary along with PNKP levels, and interpretation is therefore difficult. It would be useful to perform the NEIL2 dose-response analysis in the presence of a constant level of PNKP.

We thank the reviewer for the comment, and concur that the experiments depicted in Figure 4 as originally described and presented are difficult to understand. Therefore, we added another panel (a new version of Figure 4A) showing a schematic of the expected APE1 and NEIL2 reaction products, and hopefully this is a clearer description of what we did and found. Our original Figure 4A (Figure 4B in the revised manuscript) did not show reduced activity of APE1. Rather, lanes 5 to 7 showed respectively: the lyase products of the combined reaction of NEIL2 and APE1, NEIL2 alone, and APE1 alone on the AP sites generated by UDG. APE1 alone generates a 25-nt-3’OH terminated product (lane 7); NEIL2 generates a 25-nt-3’P terminated product, which migrates somewhat faster than 3’OH-terminated APE1 product (lane 6). When both enzymes were added, only the faster migrating produce was seen (lane 5), indicating it was generated by NEIL2 and that NEIL2 outcompetes APE1 on the single-stranded AP site (Figure 4B in the revised manuscript).

As to the reviewer’s comment “It looks like double-stranded oligo is not a suitable substrate for NEIL2. APE1 is much faster, and a higher concentration of NEIL2 can compete for only 50% activity. Figure 4C and D is very complex to understand ….“. We think that this comment reflects some misunderstanding on what we did: All of the double-stranded AP-containing substrates (generated by UDG treatment of the U/G mispair) can be digested by either APE1 or NEIL2, as shown in our original Figure 4B (Figure 4C in the revised manuscript). This figure showed concentration curves of the two enzymes, which allowed determination of the amount of either enzyme that was not quite or barely sufficient to digest all of the substrate. We then carried out a competition experiment between NEIL2 and APE1 using increasing amounts of NEIL2 at the two concentrations of APE1: the amount (0.005 U) barely sufficient to digest all of the substrates and a 4-fold higher amount (0.02 U), shown respectively in Figure 4C,D in our original submission.

To make the logic of this experiment clearer, we combined the C and D panels in the original Figure 4 into one panel (Figure 4D in the revised manuscript). Also, we labeled the lanes corresponding to the two concentrations of APE1 as “Low APE1 (0.005U)” and “High APE1 (0.02U)” respectively. Lanes 2 and 3 (Figure 4D in the revised manuscript) are positive controls to show the NEIL2-only (double band) and APE1-only (single band) products. That the distinctive double band NEIL2 product appears at about the same concentration of NEIL2 even at a 4-fold excess of APE1 strongly supports our contention that NEIL2 usurps APE1 at the AP sites. We also rephrased the relevant parts of the text to clarify the description of the Results: subsection “NEIL2 outcompetes APE1 at AP sites” and Legend to Figure 4 in the revised manuscript.

Additionally, as suggested by the reviewer, we performed the incorporation assay in the presence of a constant level of PNKP and increasing amounts of NEIL2, and the result (Figure 4F in the revised manuscript) is consistent with our previous one.

6) A substrate using THF as an AP site substitute can be used as a control. Also using a substrate with 3'P rather than UDG can be a cleaner set-up to compare.

As AP sites are probably the most common naturally occurring DNA lesions, we felt that they would be the most appropriate substrate for our in vitro studies. Our method using UDG to generate AP sites from U-containing single- or double-stranded oligonucleotides is a well-accepted standard procedure. To confirm the identify of AP sites, we treated the product from the UDG reaction with NaOH, which cleaves the AP sites upon heating at 95°C. The results showed that only the product from the UDG reaction can be cleaved by NaOH treatment (Figure 4—figure supplement 1C in the revised manuscript).

7) Although the authors stated that Exo1 activity is similar for both APE1 and NEIL2 products, Figure 5E shows different substrate to product ratios for two enzymes. Authors might consider adding a panel of APE1 and NEIL2 together in Exo1 activity assay in Figure 5E to understand if the presence of APE1 facilitates or inhibits Exo1 activity.

We thank and concur with the reviewer that we should use equal amounts of APE1 and NEIL2 products to compare the relative activity of Exo1 on these substrates. We therefore repeated this experiment with equivalent levels of NEIL2- and APE1-generated substrates, and showed that both substrates were equally digested by two different concentrations of Exo1 (Figure 5E in the revised manuscript).

[Editors' note: further revisions were requested prior to acceptance, as described below.]

The authors have addressed an important concern, providing better descriptions of the specialized methodology used in the paper. The authors have also removed some controversial data from the paper, resulting in a succinct, readable and focused report. However, some controls are still missing:1) An experimental validation of the proposed competition between APE1 and NEIL2 (usurpation of APEI by NEIL2) is critical. A ChIP-seq or a chromatin binding assay can be used for this purpose. If validated, this point is potentially very important, as is emphasized in the Abstract.

The following two points summarize the current state of how we had experimentally validated the proposed competition between APE1 and NEIL2 for abasic (AP) sites, which we had termed usurpation of APE1 by NEIL2. In our revised manuscript we repeated some of these experiments and made all the modifications suggested in the review of our original submission:

1) Our current Figures 4 and 5 unambiguously show with purified proteins that, depending on their relative concentrations, NEIL2 outcompetes (usurps) APE1 on a defined AP site, interferes with the subsequent Polβ step of base excision repair (BER) and generates a substrate susceptible to Exo1, which would generate a single-stranded A3B deaminase substrate.

2) We recapitulated the essential elements of this finding in vivo by showing that overexpressing NEIL2 in the MDA-MB-453 cell line, which has low expression of NEIL2, induces both U/G repair-mediated A3B mutations (Figure 2F) and DNA double strand breaks (Figure 3D) as measured by immunofluorescence of γH2AX foci. We stress here that while NEIL2 is biochemically unable to process a U/G site (Figure 4—figure supplement 1B), it can process and cleave the AP site generated from U/G by a U/G-specific glycosylase (Figure 4C).

The reviewer suggests that “A ChIP-seq or a chromatin binding assay can be used for this purpose”. ChIP-seq is normally used to determine sequences targeted by nucleic-binding proteins after their immunoprecipitation by a cognate antibody. In our case, the targeted sequence would be an AP site. Is the idea here to compare the relative numbers of AP sites targeted by NEIL2 and APE1 in the cancer cell lines that differ in repair-mediated mutation? Aside from the fact that there is no good NEIL2 antibody available, the AP site is a substrate for both enzymes and would presumably be eliminated by their enzymatic activity upon binding. It is infeasible for us to score such an outcome. Chromatin binding assay has been used to directly detect binding of proteins to chromatin. Is it suggested that we compare the binding of APE1 and NEIL2 to bulk chromatin? Even if we could control for the likely non-specific binding of these proteins to bulk chromatin, we do not think this experiment could supplement the unambiguous competition for AP sites that we identified using purified components (Figure 4A-D).

2) Due to the absence of appropriate measurements of protein levels (for example, NEIL2 and APE1 in nuclear extracts that show variable activity in deaminase assays) it is not possible to rule out that the observations reported in the paper might be explained by alternative mechanisms. If appropriate antibodies are not available, these other mechanisms should at least be discussed.

Given that the results presented in Figures 4 and 5 are congruent with the known elements of BER and its perturbation by NEIL2, we have proposed what seems to be the most straightforward mechanism for these results. However, the reviewer makes a good point and we have modified the relevant part of our Discussion accordingly. The relevant text now reads:

“Therefore, the most parsimonious explanation of our results is that NEIL2 diverts BER to Exo1-generation of single-stranded DNA that would be vulnerable to A3B deaminase (Figure 5F). […] The inherent reductive power of a biochemical experiment using purified components is its ability to reveal the prevailing baseline conditions of a process (Figure 5F).”

In addition, we would be pleased to consider and discuss other mechanisms that the reviewer suggests.

3) The authors are correct to suggest that immunofluorescence is an acceptable measure of DNA damage in some literature. However, because of the high signal-to-noise ratio in the current submission, the paper would still benefit from another measurement of the extent of DNA damage based on a different method, for example, immunoblotting of γH2AX vs. total H2AX.

The reviewer implied that because of the “high signal-to-noise ratio” we should consider an additional assay for γH2AX. But a “high signal-to-noise ratio” is usually considered a good feature of an assay, which is what our immunofluorescence results showed and agreed with previous assays for γH2AX foci in Hs578T cells (Morel et al., 2017). The reviewer correctly states that immunoblotting of γH2AX vs. H2AX has been used to measure the double strand breaks (Burma et al., J. Biol. Chem. 2001). However, as we wrote in our previous response to reviewers’ comment #4 in the first round of revision, quantification of γH2AX foci by immunofluorescence is the most widely used, direct and accurate way to quantify double strand breaks in cells (Morel et al., 2017; Ladstatter et al., Cell, 2016; Lobrich et al., Cell Cycle, 2010; Kinner et al., Nucleic AcidsRes. 2008), which is why we chose this method.

For the quantification of immunofluorescent signals in Figure 3 (A, right panel, C and D): what does the y-axis (% of γH2AX positive cells) refer to? Does each data point refer to a replicate experiment, or was the percentage of positive cells determined in each field in a single imaging experiment? How significant are the changes induced by the two shNEIL2 when compared to control levels?

We thank the editor and the reviewer for pointing this out. In Figure 3A, C, and D, the y-axis refers to “the percentage of cells with γH2AX foci”. Each data point is derived from a randomly selected microscopic field in two replicate experiments (Line 852-853, 863, and 868 of the figure legends of this version of the revised manuscript). The control for the two shNEIL2 (column 3 and 4 in both panels of Figure 3A) is the data shown in column 2 because NEIL2 is the only variable in this experiment (A3B-3HA was overexpressed for these three columns but not for column 1). We have added the relevant statistical analysis to the figure (Figure 3A in the revised manuscript).